# Dissection of gene expression datasets into clinically relevant interaction signatures via high-dimensional correlation maximization

Michael Grau[1,2], Georg Lenz[1,2] & Peter Lenz[3,4]*

Gene expression is controlled by many simultaneous interactions, frequently measured collectively in biology and medicine by high-throughput technologies. It is a highly challenging task to infer from these data the generating effects and cooperating genes. Here, we present an unsupervised hypothesis-generating learning concept termed signal dissection by correlation maximization (SDCM) that dissects large high-dimensional datasets into signatures. Each signature captures a particular signal pattern that was consistently observed for multiple genes and samples, likely caused by the same underlying interaction. A key difference to other methods is our flexible nonlinear signal superposition model, combined with a precise regression technique. Analyzing gene expression of diffuse large B-cell lymphoma, our method discovers previously unidentified signatures that reveal significant differences in patient survival. These signatures are more predictive than those from various methods used for comparison and robustly validate across technological platforms. This implies highly specific extraction of clinically relevant gene interactions.

---

[1] Department of Medicine A, Albert-Schweitzer Campus 1, University Hospital Münster, 48149 Münster, Germany. [2] Cluster of Excellence EXC 1003, Cells in Motion, University of Münster, 48149 Münster, Germany. [3] Department of Physics, Renthof 5, University of Marburg, 35032 Marburg, Germany. [4] LOEWE Center for Synthetic Microbiology, 35032 Marburg, Germany. *email: Peter.Lenz@physik.uni-marburg.de

The analysis of gene expression (GE) using microarrays or high-throughput RNA-sequencing[1] allows the determination of molecular interactions and GE programs in cancer cells[2,3]. However, these technologies measure concurrent GE programs in all cells of a sample collectively. They cannot directly determine which genes cooperate in specific cellular functions. It remains a major challenge to discover, dissect and extract such interactions.

Essential characteristics of cancer samples[4–14] are represented by GE signatures, i.e. by sets of cooperating genes. For diffuse large B-cell lymphoma (DLBCL)[2], the most frequent lymphoma type in adults[15], hierarchical clustering (HC) detected GE signatures that distinguish two molecular subtypes derived from different cells of origin (COO)[12]. These are clinically relevant, as patients of the germinal center B-cell-like subtype (GCB DLBCL) have a 5-year overall survival (OS) of approximately 80% compared to only 50% for patients diagnosed with the activated B-cell-like subtype (ABC DLBCL)[2]. DLBCL is a molecularly heterogeneous disease[2,16,17] and it remains a challenge to discover all disease-specific GE programs and their influence on patient outcome. Even within ABC and GCB DLBCLs, heterogeneity prevails and significant GE differences remain within these subgroups. A significantly better understanding of this heterogeneity is required to ultimately be able to treat affected patients differently.

Detecting interactions in GE datasets for many genes and samples is a highly challenging underdetermined problem. Supervised methods such as gene set enrichment analysis (GSEA)[18] use previously described information in their search for associations, while unsupervised methods aim to identify novel signatures in an unbiased manner based on GE data only. Generally, any gene subset showing significant co-expression for any subset of samples could represent a biological interaction. Detection methods therefore utilize search strategies. HC uses distance measures and linkage methods[19] to identify interacting subsets. Since its first systematic application to GE datasets[20], HC has provided many important insights[4–14,21–23], such as tumor-immune cell interactions in breast cancer[6]. Principal component analysis (PCA)[24–26] obtains principal components (PCs) by searching for maximal variance. Often few PCs already capture most of the GE variability[27–29]. Non-negative matrix factorization (NNMF)[30] minimizes a divergence functional[31] and has also identified clinically relevant tumor subtypes[32]. Biclustering methods such as FABIA[33] or PLAID[34] cluster genes and samples simultaneously and successfully rediscovered, e.g., breast cancer subclasses[33]. Independent component analysis (ICA)[35,36] maximizes statistical independence. Obtained independent components (ICs) may be interpreted as specific oncogenic pathways or regulatory modules[37].

All these methods have broad ranges of applicability, but various limitations, e.g., all require missing values to be imputed in advance. While HC can identify isolated signatures[5,8,9], it cannot dissect overlapping interactions[10,13,14]. As genes influenced by multiple interactions may show higher variability, resulting PCs may represent unspecific GE mixtures that obstruct biological understanding[25]. ICA tries to prevent such mixtures by maximizing a measure of non-Gaussianity[36], but this does not work for normally distributed data[38]. NNMF and most biclustering methods require that the unknown number of signatures is provided in advance. Additionally, NNMF is restricted to positive signals, i.e., it cannot model gene suppression.

Here, we present signal dissection by correlation maximization (SDCM) that overcomes these limitations and that significantly extends the class of detectable interaction signatures. With interaction we generically refer to any cause of correlations between arbitrary genes in arbitrary subsets of samples. Multiple interactions may affect the same genes and samples. An interaction signature aims to extract all traces of correlation in the signal that originate from one particular interaction. After introducing SDCM, we thoroughly validate our approach and systematically compare it to 17 other approaches of unsupervised learning. We apply our algorithm together with several comparison methods to real GE data from human DLBCL samples and rank all results based on their ability to reveal differences in patient survival.

## Results

**SDCM concepts**. The original idea of SDCM is a unifying graphical model for biologically relevant GE data that is typically represented by a heatmap depicting gene and sample dimensions simultaneously (such as Fig. 3 in ref. [3]). Our model assumes that an individual GE program is associated with specific orders of genes and samples. In absence of any overlapping effects and measurement noise, we model GE data that was sorted by such specific gene and sample orders as a heatmap in which each participating gene row and each sample column is comprised of monotonic expression values only, i.e., all follow a common order. This consistency idea is the key for guiding data regression and dissecting overlapping GE programs that were measured simultaneously as a sum of all contributing intensities. SDCM dissects the complete GE dataset into such bi-monotonic signatures for specific gene and sample orders. Despite empowering dissection, this non-linear consistency model also significantly increased versatility and scope of detectable interaction signatures compared to linear methods.

To illustrate SDCM concepts, we first show a low-dimensional gene space and compare our approach with PCA. PCA searches for directions of maximal data variance. Quantification of data variance in any given direction ignores perpendicular distances of data points. In contrast, our search functional is maximal for directions, to which as many data points as possible are aligned as consistently as possible. To quantify this, we compute uncentered weighted correlations of data points with candidate directions (aka weighted cosine similarities). Weights decrease with angular distance to that direction. Geometrically, these weights enable SDCM to focus on data points in a double-cone around any given direction (with cone tips touching at origin, as illustrated in Fig. 1c–f). While a PC represents a data point distribution as a linear axis, SDCM extends this linear concept to nonlinear monotonic curves obtained by regression. Points with relatively low weights (outside of the cone) have lower influence on this regression than data points in the cone. In this way, SDCM extracts data structures locally, whereas PCA globally reduces data dimension by projection along the PC of maximal variance. This can result in a 1-to-many relationship between GE program and PCs, unnecessarily obstructing biological insight. We illustrate this problem in our 3D concept example that contains four simulated GE programs for different partitions of samples. For this dataset, the first PC of maximal variance passes through the empty space between the three larger simulated GE programs (Fig. 1i). As PCs are orthogonal per construction by projection, there are only three PCs in total for this 3D gene space. Hence, the optimal 1:1 relationship with the four simulated programs can principally not be obtained with PCs. The same is true for any other method utilizing projections for dimension reduction, e.g., ICA.

This problem is aggravated in high-dimensional data spaces, as more genes could play multiple roles in GE programs of different patient subsets. These subsets are then hard to represent by orthogonal PCs, similar to the 3D example. Furthermore, in high-dimensional biological data, samples usually harbor multiple GE

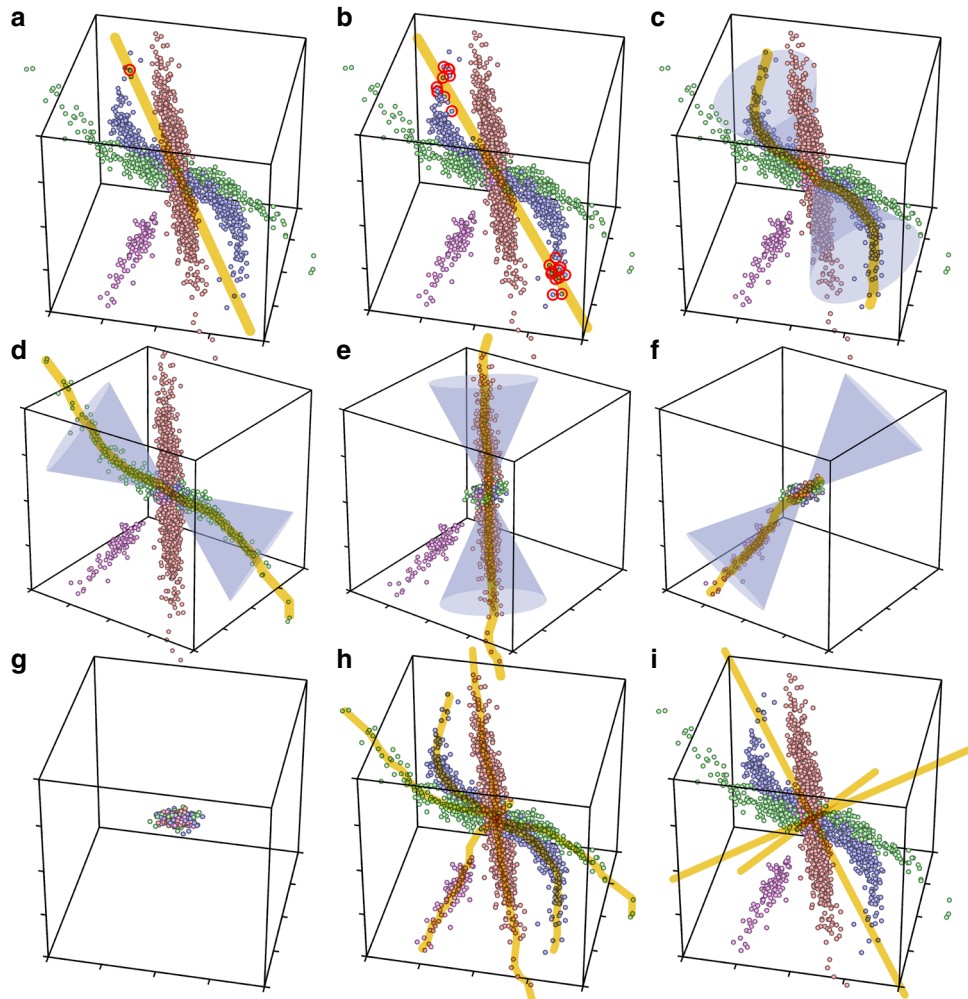

**Fig. 1** Concepts of SDCM illustrated by a 3-dimensional example. Four interactions indicated by different colors are simulated in a space spanned by three genes. Points represent simulated $\log_2$(gene expression ratios) and are analyzed by SDCM (without color information). **a** A first representative of the simulated blue interaction is identified by the search strategy, defining the initial gene axis (yellow; Methods/Step 1). **b** Additional representatives (red) are selected that maximize the signature functional (Methods/Step 2). The converged gene axis (yellow) approximates the direction of interaction by a linear combination (like a principal component). **c** The signal of the blue interaction is regressed according to our bi-monotonic signature model. Regression focuses on a local subspace using weights decreasing by angular distance, as illustrated by cones (depicting isosurfaces for uncentered unweighted correlations of $[\mathbf{x}|\mathbf{a}]_{|1)} := 0.9$). The resulting signature signal (illustrated by its corresponding gene curve, yellow) approximates the interaction more precisely (Methods/Step 3). **d** The discovered blue interaction (implemented as a logistic function simulating a saturation) is dissected without disturbing the signal from other interactions (Methods/Step 4). In the second detection iteration, steps a-c are repeated, resulting in the regressed gene curve for the green interaction (a super-linear polynomial). **e** After dissection of the green interaction, the (linear) red interaction is detected. **f** After dissection of the red interaction, the magenta interaction (simulating a one-sided activation threshold) is detected. **g** The residual signal no longer contains any qualifying signature representative and SDCM terminates. **h** All four detected signature signals are shown in form of their corresponding gene curves (yellow). **i** All three orthogonal principal components are depicted (yellow), as obtained by PCA for the same data points.

programs that are often overlapping (sharing genes). Hence, real GE datasets usually do not give rise to an unambiguous patient partitioning as simulated for the 3D example. Due to these complexities of real GE data, we shifted goal from dimension reduction, as pursued by PCA, to consistent 1:1 representation of initially overlapping interaction signatures. Similar to biclustering methods, SDCM works in full data space, treating gene and sample space on equal footing, and rather than reducing dimensions, the data signal itself is iteratively reduced to a noise cloud around zero (cf. Fig. 1g).

Because of these significant conceptual differences we had to mathematically develop SDCM from scratch based on a general superposition approach $\mathbf{M}_0 = \sum_k \mathbf{E}_k + \boldsymbol{\eta}$. Here, the input signal, represented as a genes*samples matrix $\mathbf{M}_0$, is dissected into a sum of signature matrices $\mathbf{E}_k$ of the same size, additively overlaid with

a noise matrix $\boldsymbol{\eta}$. While each signature $\mathbf{E}_k$ formally has the same size as the input, it explains non-zero signal parts only for typically small subsets of associated genes and samples. Noise is implicitly defined by adjustable significance thresholds for signal strengths (projections of data points onto a candidate direction) and for uncentered weighted correlations (angular distances of data points to a candidate direction). As long as data structures remain in the signal that are significant with respect to these conditions, SDCM continues to dissect signatures by iterating the following four steps:

(1) search for an initial candidate direction using a signature functional,

(2) optimize this direction by locally maximizing the signature functional,

$$\mathbf{M}_0(\mathbb{I}_k, \mathbb{J}_k) - \sum_{l=1}^{k-1} \mathbf{E}_l(\mathbb{I}_k, \mathbb{J}_k) = \mathbf{M}_{k-1}(\mathbb{I}_k, \mathbb{J}_k) = \mathbf{E}_k(\mathbb{I}_k, \mathbb{J}_k) + \mathbf{M}_k(\mathbb{I}_k, \mathbb{J}_k)$$

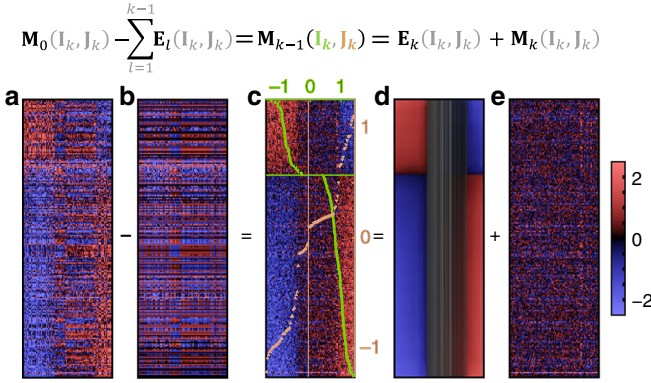

**Fig. 2** Dissection iteration for a high-dimensional signal. A high-dimensional signature is detected and dissected from our versatility test scenario containing 13 superposed signatures and noise. **a** The input signal $\mathbf{M}_0$ is shown in the detected signature order. **b** Five overlapping signatures have already been detected and dissected in previous iterations. Their superposition depicts already explained parts of the signal. **c** Current signal $\mathbf{M}_{k-1}$, displayed in the detected gene order $\mathbb{I}_k$ and sample order $\mathbb{J}_k$. Sample strengths for each column (light brown) defining $\mathbb{J}_k$ were determined by projections of samples on the detected gene axis. Gene strengths for each row (light green) defining $\mathbb{I}_k$ were determined analogously. Samples sorted to the center are not affected by this signature and have strengths close to zero. **d** Specific contributions of the detected signature to the overall signal, as determined by regression according to our signature model. Consequently, these contributions are bi-monotonic in the displayed signature order ($\mathbb{I}_k$, $\mathbb{J}_k$). They are equivalent to a monotonic curve in the high-dimensional gene space. Gray shadings depict low gene and sample weights. (For a better overview, non-participating genes with weak strengths and low weights are hidden; their signature signal is close to zero.) **e** The residual signal $\mathbf{M}_k$ that remains after dissection of the regressed signature signal $\mathbf{E}_k$ is the input for the next detection iteration $k+1$. Besides simulated noise, it still contains signals from seven superposed signatures with yet undetected random gene and sample orders.

(3) regress a monotonic curve through data points in the weights-based cone and
(4) selectively dissect the signal parts that are consistent with this curve.

All these steps are illustrated for the 3D example (Fig. 1) and described in the section Conceptual introduction in Methods. Identical steps are performed in case of high-dimensional signals, as illustrated in Fig. 2. We provide definitions of algorithmic steps and concepts in the Methods section. Supplementary Notes 1–7 detail subroutines. In particular we explain two key concepts, the signature functional (that guides the initial search of a linear axis and its optimization) and our bimonotonic consistency requirements (that guide signature regression onto a nonlinear monotonic curve and subsequent signal dissection).

**Versatility test and method validation**. To validate detection versatility for high-dimensional data, we designed seven signatures $\mathbf{E}_k^{\mathrm{sim}}$ of different size and form that mimic real-world GE signatures (Supplementary Table 1). Each was simulated bi-monotonically in a random gene and sample order (Fig. 3a). The test signal $\mathbf{M}_0^{\mathrm{sim}}$ was defined as superposition $\sum_{k=1}^7 \mathbf{E}_k^{\mathrm{sim}}$ of all unordered signatures and normal noise (Fig. 3b). The challenge was to precisely detect and dissect all seven signatures and to avoid detection of false positive (FP) signatures. SDCM detected all seven signatures, reconstructed most of their signals (Fig. 3c) and stopped when the remaining signal (Fig. 3d) did no longer contain any further qualifying signature representatives. In particular, it did not return any FPs. To quantify reconstruction quality, simulated signature axes were correlated with detected ones (Fig. 3e and Supplementary Note 8). SDCM extracted all simulated signatures with correlations close to 1 (high sensitivity, red diagonals). Additionally, there was no strong correlation of detected signatures to other simulated signatures (high specificity, dark off-diagonals). These results are representative for 49 repetitions (Supplementary Fig. 1).

**Method comparisons with simulated data**. We applied HC, NNMF, FABIA, FABIAS, ICA, and PCA to the same versatility tests (Supplementary Figs. 2–6 and Supplementary Note 9). Only PCA performed comparably well. HC found only signature #1 and ICA, FABIA, FABIAS and NNMF found only a few. The latter three also mixed signals from multiple signatures (red off-diagonals). For the 3D test, neither PCA (Fig. 1i), nor HC or FABIAS (Supplementary Figs. 7 and 8) were able to recover simulated interactions (ICA is restricted to three components like PCA and FABIAS; NNMF is only applicable to positive values). To test the scope of our signature model, we applied SDCM to four external benchmarks that were previously defined to rank 13 biclustering methods[33]. Each benchmark consists of 100 datasets, one was generated with a multiplicative data model, the other three with an additive data model for different signal-to-noise ratios[33]. SDCM scored best in all four benchmarks, followed by FABIAS (Supplementary Table 2). For an in-depth comparison with PCA, we designed a more challenging test signal containing two additional instances of signatures #2, #3, and #4 from the above 7-signature test (for different random subsets and orders of genes and samples). 49 simulations of this versatility test with 13 superposed signatures were analyzed with both SDCM and PCA. Detected gene axes (respectively PCs) were correlated with simulated gene axes (Fig. 4b and Supplementary Fig. 9). SDCM was significantly more sensitive than PCA for 10/13 signatures and significantly more specific for 12/13 signatures (Supplementary Table 3).

**Number of signatures**. In contrast to the other methods, SDCM can infer the typically unknown number of signatures (termination rules in Methods/Step 1 and Supplementary Note 4). Defining FPs as detected signatures with correlations <0.4 to all simulated signatures, 53% of the 49 13-signature tests had zero FPs, 33% had one, 10% had two, 4% had three and none had ≥4 FPs (overall false discovery rate 0.048). Defining false negatives as simulated signatures with correlations <0.4 to all detected

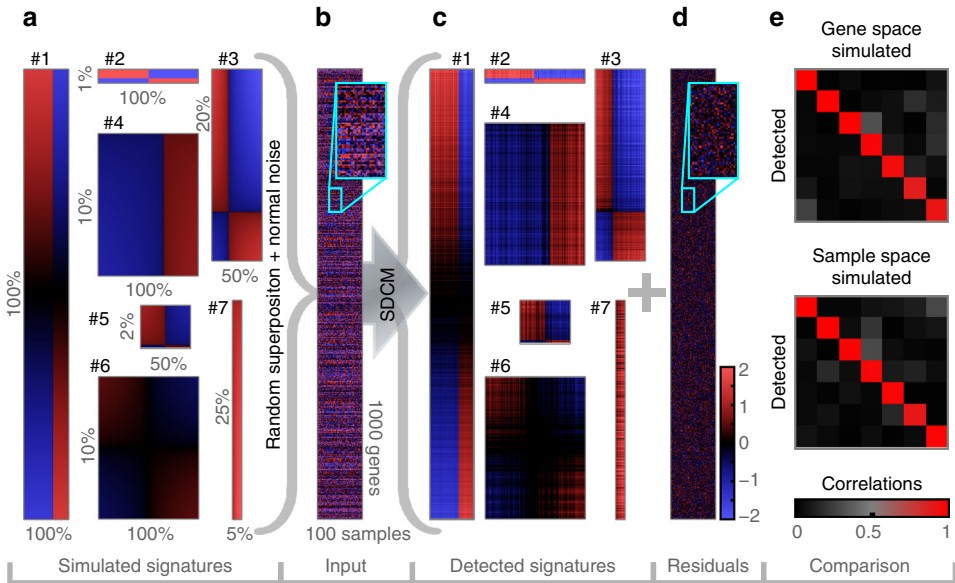

**Fig. 3** Versatility test scenario for method validation. **a** Seven signatures are simulated as depicted, representing various interactions or laboratory effects (definitions in Supplementary Table 1). To test dissection of dominating signatures, #1 was chosen to mimic two different experimental protocols affecting all genes. #2 simulates a strong binary cluster for only 1% of all measured genes; it could represent gender specific differences. To test detection of signatures that only affect sample subclasses, #3 mimics a gradual interaction that only exists in 50% of all samples. #4 simulates a one-sided interaction that is typical for disease-specific real world signatures like the stromal-1 signature[2]. To test detection specificity for small signatures, #5 contains 20 genes with only one being anti-correlated. To test detection sensitivity with respect to noise, #6 simulates a weak signal. Finally, #7 tests sensitivity for narrow expression offsets in only 5% of all samples. **b** Signals for all seven signatures are superposed for random gene and sample orders. Normal noise is added (SD = 0.5). The resulting signal (displayed in an arbitrary gene and sample reference order) is input to signal dissection by correlation maximization (SDCM). **c** All seven signatures and zero false positives were detected. Detected signature signals are displayed in their known simulated gene and sample orders for a direct visual comparison with (**a**). All major gradual changes were reconstructed. **d** Residual signal after seven dissection iterations. **e** Similarities between simulated and detected signatures (columns represent simulated gene or sample axes for patterns #1–#7, rows correspond to detected axes in best-matching order; formulae in Supplementary Note 8). Red diagonals indicate high detection sensitivity. Black off-diagonals indicate high detection specificity.

signatures, there was only 1/49 simulations with 1/13 signatures that was not detected (overall sensitivity 0.998; Supplementary Fig. 9).

**Limits**. To determine the minimal signal strength required for detection, we simulated single instances of signature #6 for decreasing maximum signal strengths relative to the noise level σ. SDCM detected this signature down to approximately 0.5σ. Below 0.5σ, SDCM often terminated without any detected signature (Supplementary Fig. 10). No FP signatures were returned. For correlations $r \geq 0.8$ to the simulated gene axis (indicating good reproducibility), SDCM required approximately 0.75σ. Here, PCA performed better by requiring only 0.63σ. PCA always returns PCs. However, for weak strengths ≤0.4σ, all were FPs ($r \leq 0.2$). Next, we asked how many interactions may overlap while still being dissectible. We simulated test signals with an increasing number of superposed signatures (plus noise). For series based on signatures #3 or #4, SDCM showed high detection rates. The series for the weak signature #6 illustrates superposition limits of our method. PCA performance broke down completely for all three superposition series (Fig. 4c and Supplementary Figs 11–13). Scaling limits of SDCM are determined by its analytic complexity $O(\hat{k}(m+n)nm)$ (see Eq. 37). Consistently, measured computation times scaled quadratically in $m$ (Supplementary Fig. 14). A typical runtime is 5 min per signature ($m = 20000, n = 100$).

**Missing values and their reconstruction**. Various measurement problems or quality filters can lead to missing values. The other methods require either exclusion of affected genes or replacement values. As regression in SDCM is based on weights (see the section Signature focus ($|\mathbf{w}^g\rangle, |\mathbf{w}^s\rangle$) in Methods), we support missing values in a natural way by assigning zero weights. SDCM additionally predicts specific contributions from missing values to each detected signature based on our signature model. For the 7-signature versatility test, SDCM detected all signatures with high correlations up to missing value rates of 35% (Supplementary Fig. 15a). Even for 80% missing values (Fig. 4d), signatures #1–#4 were still detected with high correlations and the majority of their missing values were reconstructed (Supplementary Fig. 15b).

**Application of SDCM to GE data from DLBCL samples**. For signature discovery in real data, we selected the largest GE dataset of human DLBCL samples available at time of this analysis with 498 patients[3] (Supplementary Note 9). SDCM dissected this dataset into 105 signatures (Supplementary Data 1 and 2). For direct usage of discovered signatures in a supervised analysis such as GSEA[18], we also provide derived sets of signature top genes (Supplementary Data 3). Real datasets typically contain cohort-specific laboratory effects. While SDCM dissects these from biological signals, resulting FP signatures have to be filtered out.

To verify signatures on GE level and test their robustness after changing the technological platform, we transferred them to both a microarray-based validation cohort of 233 independent DLBCL patient samples[2] and to a RNA-sequencing based validation cohort of 624 independent DLBCL patient samples[39] (Supplementary Note 11). To this end, only gene axis and gene weights of each signature are transferred. Projections of target samples onto transferred axes then yield signature-specific orderings of the

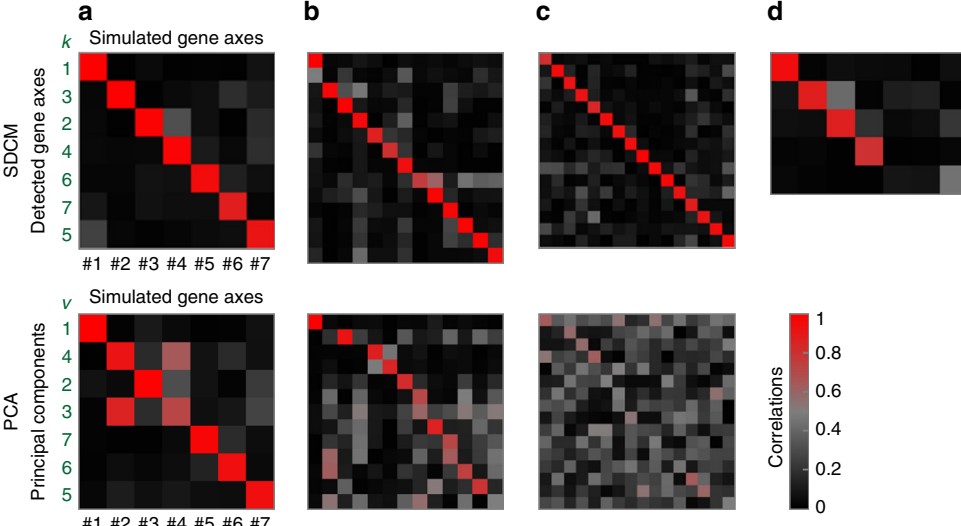

**Fig. 4** Detailed comparison with PCA. Detection performance of SDCM (top) and PCA (bottom) for different test scenarios. Colors indicate correlations of detected gene axes (respectively PCs) to simulated ones (Supplementary Note 8). Each matrix column represents a simulated signature; each row a detected one. Red diagonals indicate high detection sensitivity, dark off-diagonals high specificity. Bright pixels in the same row indicate a detected gene axis (or PC) that represents a mixture of multiple simulated signatures. Bright pixels in the same column indicate a simulated signature signal that was split over several detected ones. **a** Representative results from SDCM and PCA for the 7-signature versatility test (from Fig. 3; all runs in Supplementary Figs. 1 and 6). We present SDCM gene axes respectively PCs in best-matching order. SDCM detected all seven signatures and no FPs. PCA yielded 100 PCs; the top 7 by variance are shown. PCA performed quite well but mixed signatures #2 and #4. **b** Representative results for the 13-signature versatility test (all runs in Supplementary Fig. 9). SDCM yielded one FP and slightly mixed signatures #8 and #9. Otherwise, it retained high detection specificity. The top 13 PCs showed considerably more mixing; only signature #1 retained high specificity. **c** Representative results for the superposition test with 16 instances of signature #3 (from Fig. 3a; complete series in Supplementary Fig. 11). SDCM maintained high detection performance, while PCA's sensitivity and specificity broke down. **d** For a versatility test as in (**a**), but after random deletions resulting in 80% missing values (Supplementary Fig. 15), SDCM still detected 4/ 7 signatures with high sensitivity. (PCA does not support signals containing missing values).

target cohort. Hence, signatures are validated on GE level if the target cohort exhibits similar significant differences between samples as in the source cohort, resulting in similar correlations between genes (Supplementary Fig. 16 shows signature $k = 6$ as example). Additionally, strongly bimodal sample strengths in signature $k = 6$ induced binary clusters that were significantly associated with independent patient gender data (detection cohort: $p = 2.8 \times 10^{-83}$, microarray-based validation cohort: $p = 5.7 \times 10^{-50}$, RNA-sequencing based validation cohort: $p = 6.2 \times 10^{-98}$, $\chi^2$-tests). Albeit as disease-unspecific as gender, this signature is a first biological proof of concept and it may also serve as a control signature for future dissections of human GE studies.

To filter signatures, they were associated with previously described disease-specific gene signatures or clinical data. Interpretations of these associations for all discovered signatures are beyond the scope of this study. Here, we focus on those of key importance for method validation. In the study by Lenz et al.[2], three GE signatures were identified and combined into a survival predictor model. This model was able to identify patients with significant differences in survival. SDCM rediscovered all three signatures, as confirmed by high and significant gene set enrichment[18] (Supplementary Figs. 17–19 and Supplementary Note 12). In particular, we refer to iteration $k = 12$ as rediscovered COO[40] signature, as it showed the highest enrichment for known ABC and GCB DLBCL gene sets. This confirms the utility of SDCM in discovering cancer subclasses. Size, inner correlation and amount of explained signal for all 105 discovered signatures are depicted in Supplementary Fig. 20. Signatures detected in early iterations already explain most of the signal, while at later iterations, signatures are typically smaller (fewer genes in their Signature focus ($|\mathbf{w}^g\rangle, |\mathbf{w}^s\rangle$), see Methods).

As general statistics like signature size or the amount of explained signal cannot robustly indicate biological relevance, we systematically tested the association of every signature with patient survival data (see below).

**Method comparison with real data**. We also applied PCA, ICA, NNMF and FABIA/S to this real dataset (same 13 configurations as tested for simulated data). We compared detected gene axes (PCs, ICs, NNMF or FABIA/S loading vectors) with SDCM signatures via weighted correlations (Supplementary Note 8). Results are shown in Supplementary Figs. 21–24 and summarized in Supplementary Note 13. In brief, while the strong gender-associated signature and the large Stromal-1 signature were rediscovered by several configurations, none of the comparison methods rediscovered the top survival signatures identified in this study with high correlation. Additionally, artefacts were revealed, such as splitting the signal of the gender-associated signature over multiple components. Overall, ICA results showed the highest similarity to SDCM.

**Survival models and method comparison**. To determine in an unbiased fashion which of the signatures are potentially responsible for patient survival differences, and which detection method extracted the most predictive signatures, we constructed alternative Cox proportional hazard models (CPHM)[41]. We modeled the progression-free survival (PFS) of patients with combinations of up to three signatures from any of the detection methods (Supplementary Note 14 and Supplementary Fig. 25). We tested every combination of two or three signatures discovered by SDCM (including rediscovered stromal signatures) or by any of the compared methods. The best 2-signature model was

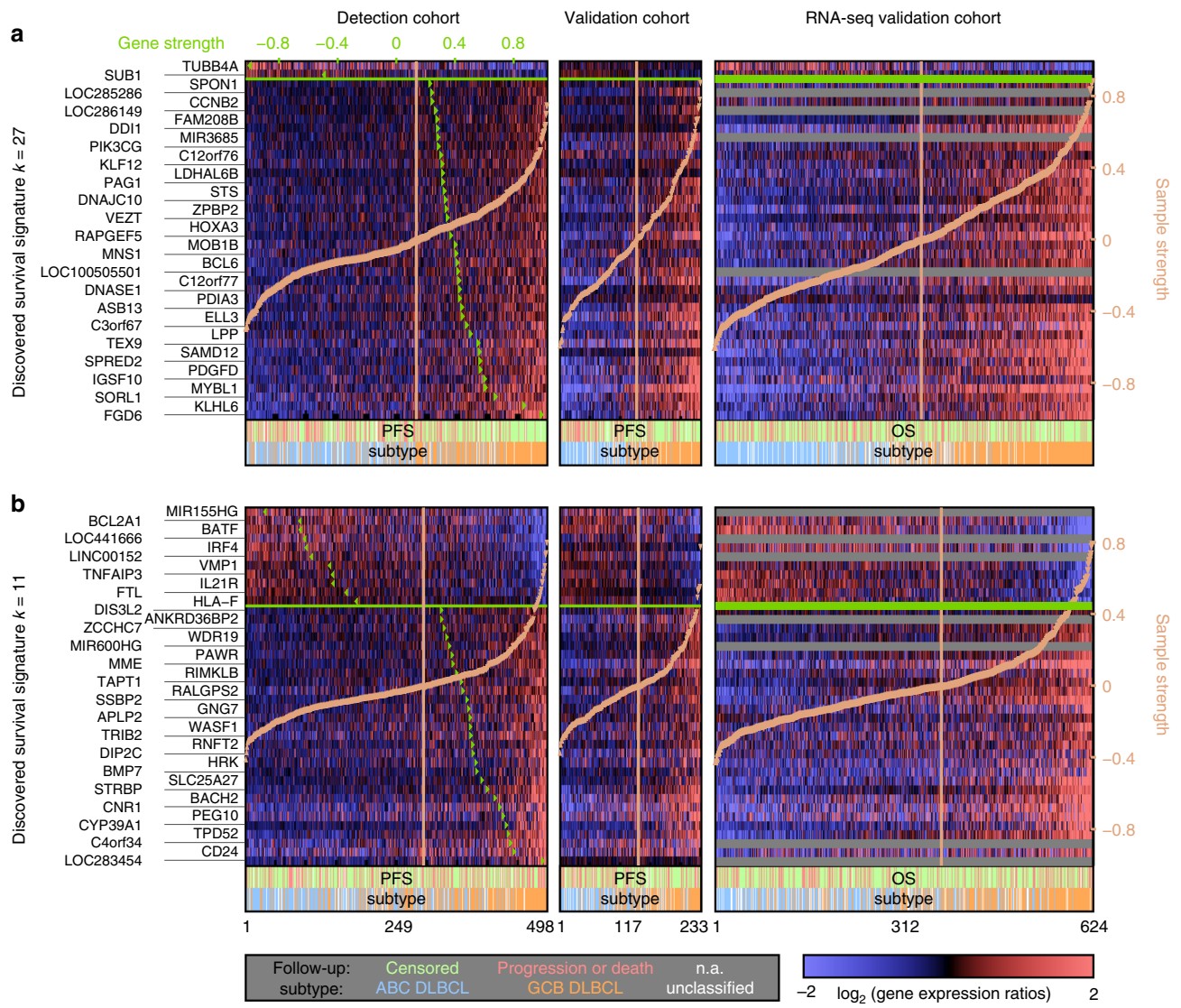

**Fig. 5** Discovered survival signatures in DLBCL. **a** The top survival-associated GE signature was detected by SDCM in iteration $k = 27$. Depicted are $\log_2$(ratios) for the top 40 genes by gene weights $|\mathbf{v}^g|$ (Eq. 21), ordered by their signature-specific strengths $\left|\mathbf{u}^g_{k,\hat{o}}\right|$ (Eq. 29; green triangles in rows). Samples are ordered by their signature-specific sample strengths (light orange triangles in columns). Heatmaps show GE intensities in the detection cohort (left), the microarray-based validation cohort (center) and the RNA-sequencing based validation cohort (right). Colored lines below heatmaps indicate outcome (light red; PFS available for microarray-based cohorts; OS for RNA-sequencing) and subtype. In all cohorts, ABC DLBCL cases (light blue) are more frequent at negative sample strengths, while better outcome occurs for higher sample strengths. Signature details on transcript cluster level are presented in Supplementary Fig. 26. Gray lines indicate missing matches between platforms. Tabular definitions (signature axes and strengths for microarray-based cohorts) are provided in Supplementary Data 1 and 2. **b** The second survival-associated GE signature in the best predictor model was detected in iteration $k = 11$. As in (**a**), its top 40 genes are displayed. Similar to (**a**) and in all cohorts, colored lines indicate a higher frequency of GCB DLBCL cases (orange) at higher sample strengths. Other than in (**a**), these are paired with an untypically high rate of progressions or deaths for this subtype (light red). (Signature details on transcript cluster level in Supplementary Fig. 27; gray lines as in (**a**); tabular definitions in Supplementary Data 1 and 2).

comprised of SDCM signatures $k = 11$ and $k = 27$. The second-best model only had a relative likelihood of 0.02 (Supplementary Table 4a). The best 3-signature model was comprised of the same two signatures and the 54th IC detected by ICA with the Gaussian contrast function[35] (Table 4b). All three top survival signatures were present in the independent validation cohort (Supplementary Figs. 26–28). Due to better predictor generalization properties in this validation cohort, we selected the simpler 2-signature model for further analysis (Supplementary Figs 29–30). Comprising survival signatures are presented in Fig. 5. Corresponding $p$-values in the final Cox model were $p_{\text{CPHM}, k=27} = 2.1 \times 10^{-11}$

and $p_{\text{CPHM}, k=11} = 2.2 \times 10^{-9}$ (details including Cox $\beta$ coefficients provided in Supplementary Note 15).

Signatures $k = 27$ and $k = 11$ represent overlapping opposite survival associations (fitted Cox coefficients have different signs). Signature $k = 11$ did not seem to be predictive when tested alone ($q_{\text{LRT}} = 0.47$, likelihood ratio test on top of the age-only model, Bonferroni-corrected for the SDCM signature family). Neither was it predictive on top of the rediscovered COO signature $k = 12$ ($q_{\text{LRT}} = 0.30$). However, prediction was highly synergistic when combined with signature $k = 27$ ($q_{\text{LRT}} = 6.8 \times 10^{-6}$). Kaplan–Meier survival estimates for four molecular risk groups visualize

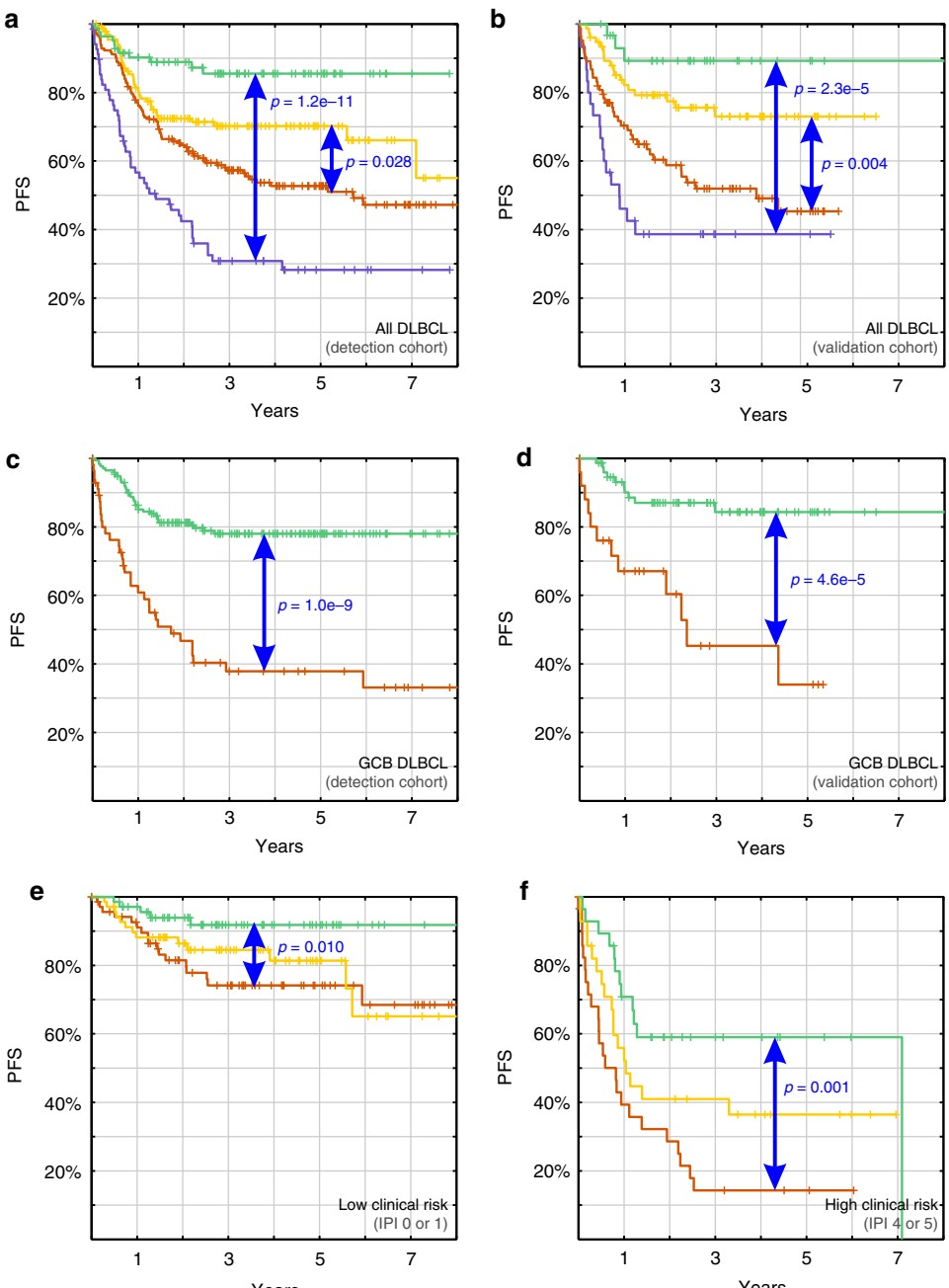

**Fig. 6** Survival differences within DLBCL. **a**, **b** To visualize the survival trend revealed by our continuous 2-signature predictor, we show Kaplan-Meier estimates for partitions of predicted molecular risk (of ≥175%, ≥100%, ≥1/175%, and <1/175% the average risk in the detection cohort). Significant differences in progression-free survival were predicted in both the detection cohort (**a**, 470 R-CHOP treated DLBCL patients) and the validation cohort (**b**, 220 patients). (Statistics of each survival curve and OS can be found in Supplementary Fig. 30.) **c**, **d** Within GCB DLBCL (as originally classified), the quartile containing highest predicted molecular risks revealed significantly and strongly adverse survival, whereas the other three quartiles shared average survival (merged into one curve; details and ABC DLBCL are depicted in Supplementary Fig. 32; OS and second validation cohort in Supplementary Fig. 33). **e**, **f** To analyze and visualize the relationship between molecular risks and known clinical risks, we repartitioned all samples having an international prognostic index (IPI)[42] into terciles. In each clinical risk class, significant survival differences remained between top and bottom terciles of predicted molecular risks (details in Supplementary Fig. 35).

our predictor (Fig. 6a). Between high molecular risk (patients with ≥175% of the average risk in the detection cohort) and low molecular risk, a difference in 5-year PFS of 58% was detectable ($p = 1.2 \times 10^{-11}$ via log-rank test reasserts the Cox model; Supplementary Fig. 30c shows OS). In the smaller validation cohort, the difference in 5-year PFS was still 53% for identical molecular risk groups (Fig. 6b, $p = 2.3 \times 10^{-5}$; Supplementary Fig. 30d shows OS). To further confirm robustness after changing the

technological platform, we applied our predictor to OS in the independent RNA-sequencing validation cohort (unfortunately no PFS available). While expected cohort-to-cohort variations were observed, strong survival differences in 5-year OS of 40% remained between identical molecular risk groups ($p = 4.9 \times 10^{-7}$; Supplementary Fig. 30e).

Collectively, this indicates that identified signatures represent specific characteristics of DLBCL and not just of a single DLBCL

cohort. In contrast, survival differences between previously described DLBCL subtypes did not generalize as well across these cohorts (Supplementary Fig. 31).

Furthermore, the identified 2-signature model discriminated patient survival significantly better than the 3-signature model of Lenz et al.[2] ($p_{LRT} = 1.3 \times 10^{-8}$, likelihood ratio test on top of previous predictor scores, Supplementary Note 16). Vice versa, the former 3-signature predictor could not add significant predictive information on top of our 2-signature model ($p_{LRT} = 0.07$).

Applied separately to DLBCL subtypes (Supplementary Fig. 32), our predictor revealed a subgroup within GCB DLBCL that was associated with significantly adverse survival ($p = 1.0 \times 10^{-9}$, log-rank test, Fig. 6c). This subgroup was confirmed in both the smaller cohort ($p = 4.6 \times 10^{-5}$, Fig. 6d) and for OS in the RNA-sequencing validation cohort ($p = 0.001$ after crossing cohort, technology and survival type; OS results for all cohorts in Supplementary Fig. 33). Sample strengths in both survival signatures showed that ABC DLBCL samples cluster densely at lower strengths in both signatures. In contrast, GCB DLBCL cases expressed these signatures heterogeneously and were scattered from high to low risk (Supplementary Fig. 34). To quantify the information provided by our molecular predictor on top of known clinical risk factors, we grouped samples by their international prognostic index (IPI)[42] that combines the clinical parameters performance status, age, stage, number of extranodal sites and serum LDH level. Interestingly, predicted survival differences remained significant in all clinical risk groups (Fig. 6e, f and Supplementary Fig. 35).

**Biological associations**. To elucidate biological associations of the two top survival signatures in an unbiased manner, we analyzed their gene ranks by discovered gene strengths for enrichment[18] of 16.513 previously identified GE signatures[40,43,44] (Supplementary Note 12). Interestingly, both correlated and anti-correlated genes of $k = 11$ were significantly enriched with signatures differentiating Burkitt lymphoma (BL) from DLBCL samples in a supervised analysis[45] (HUMMEL_BURKITTS_LYMPHOMA_DN with enrichment score es = $-0.903$; HUMMEL_BURKITTS_LYMPHOMA_UP with es = $0.741$; each with $p \leq 0.001$ by permutation tests). In all DLBCL cohorts, correlated samples (sorted to the right in Fig. 5b) showed a BL-like GE pattern. As expected, both $k = 27$ and $k = 11$ are associated with germinal center signatures (e.g., Germinal_center_Bcell_DLBCL; es = $0.76$ for $k = 27$ and es = $0.55$ for $k = 11$; $p \leq 0.001$). Likewise, both are associated with a signature determined in an analysis based on survival data[2] (Germinal_center_B_cell_DLBCL_survival_predictor; es = $0.78$ for $k = 27$ and es = $0.63$ for $k = 11$; $p \leq 0.001$). In the $k = 11$ signature, we also detected various previously identified target genes of the oncogenic nuclear factor-kappa B (NF-$\kappa$B) signaling pathway such as *BCL2A1*, *IRF4*, *TNFAIP3* or *IL21R*. This observation was confirmed in the unbiased GSEA, as anti-correlated genes of $k = 11$ (upper side in Fig. 5b) were significantly enriched with a previously identified NF-$\kappa$B signature (NFkB_Up_bothOCILy3andLy10; es = $0.68$; $p \leq 0.001$). In contrast, $k = 27$ was not associated with NF-$\kappa$B signaling (es = $-0.25$; enrichment results are shown in Supplementary Fig. 36).

## Discussion

We provide an unsupervised learning concept for dissection of large datasets that are typically comprised of the collective signal from many unknown interactions and substantial measurement noise. As we have demonstrated, SDCM is able to discover and extract interaction signatures from such data with high sensitivity and specificity. In cancer transcriptomics, this superior

performance will most likely lead to more precise dissections of entities that are molecularly heterogeneous, allowing a more specific identification of potential oncogenic programs. However, our method is generally applicable, as underlined by its top rank among 13 biclustering methods in four external benchmark datasets. Furthermore, SDCM reliably dissected highly overlapping interactions, determined the unknown number of signatures with low false discovery rates and reconstructed missing values. Finally, using outcome as independent indicator, two SDCM signatures for human DLBCL GE data showed the highest impact on patient survival of all signatures from all compared methods, suggesting high biological signature specificity. This supports the design of required comprehensive functional validation experiments to biologically identify driving interactions of these survival differences.

A central reason for high specificity and for the broad applicability of SDCM is our flexible signature model. While PCs and ICs approximate interactions only linearly, detected $\mathbf{E}_k$ are equivalent to non-linear monotonic curves (Fig. 1c, h). They can precisely regress a much broader class of interaction signatures. Precision is crucial, as residuals missed during dissection would lead to FPs or artificially disturb not yet dissected signatures from overlapping interactions. Another reason is our signature functional $\mathcal{E}$ that guides detection and optimization. It is the higher the more genes and samples take part in an interaction and the more consistent correlations this interaction causes across participating genes and samples. This optimizes signature specificity. In contrast, PCA maximizes the data variance explained by each PC, instead of correlations to it. Resulting PCs therefore represented mixtures of several interactions in superposition tests and in real data (cf. Supplementary Fig. 21b, c). An additional key advantage compared to projection-based methods (including PCA and ICA) is that axes of different signatures are not constrained to be orthogonal. This enabled precise dissections even of partially correlated interactions (as illustrated in Fig. 1), whereas orthogonality forced them to be represented by several unspecific components (Fig. 1i). Similar artifacts were observed for NNMF and FABIA/S when dissecting real GE data (Supplementary Figs. 23 and 24). As we use a perturbative-like approach (first searching a linear signature axis and then regressing a bimonotonic signature curve), non-linearities that deviate from linearity too strongly cannot be described by bimonotonic signatures. The same is true for all comparison methods. However, we did not find any biologically meaningful pattern of this type in the context of GE data.

While our primary goal was method validation against real data, dissection of DLBCL GE data into 105 signatures has provided important insights into the molecular heterogeneity of this diagnostic category. Already the gender-associated control signature and rediscovered stromal DLBCL signatures validate the utility of SDCM. For comparison, we additionally obtained signatures using PCA, ICA, NNMF and FABIA/S for the same data. To determine which of these signatures have the largest impact on patient survival, we constructed Cox proportional hazard models and ranked them using the Akaike information criterion. The best bivariate model was comprised of two SDCM signatures. It revealed significantly higher differences in patient survival than the previously described 3-signature predictor by Lenz et al.[2] that included two stromal signatures. Instead of utilizing limited survival data directly to find associated genes by supervised learning, which may cause a high false discovery rate, we dissected GE data in an unsupervised manner first. We also validated both signatures on GE level before using any survival data. Consequently, their strong survival associations (Fig. 6a, b) in both the detection and validation cohort are independent validations, indicating

biological signature specificity and relevance. This was underlined on both GE and survival level in the RNA-sequencing based validation cohort that only became available after method development, signature dissection and predictor construction.

While both signatures sorted samples roughly from ABC to GCB DLBCL over increasing sample strengths (Fig. 5), they were associated with overlapping anti-aligned survival trends and their combination predicted survival highly synergistically. As such anti-aligned survival trends can neutralize each other, they are hard to detect by supervised analyses. They revealed a 45% difference in 5-year PFS for patients with DLBCL currently classified with a high clinical risk (Fig. 6f) and identified a subgroup within GCB DLBCL patients with more than 40% inferior 5-year PFS (Fig. 6c, d). Further analyses revealed that this subgroup is characterized by a Burkitt lymphoma like GE profile and low expression of target genes of the oncogenic NF-$\kappa$B pathway. This might potentially link to the just recently identified molecular high-grade B-cell lymphoma (MHG) subtype[46] that shares these survival and BL GE characteristics. Collectively, this confirms that SDCM is able to discover novel and clinically relevant subtypes, without using prior knowledge of GE signatures or survival data. Remaining GCB DLBCLs with relative high expression of NF-$\kappa$B (left in Fig. 5b) consequently showed superior survival relative to average GCB DLBCL patients. This subgroup might be linked to a previously identified GCB DLBCL subtype with favorable outcome that is driven by BCL2 and active NF-$\kappa$B signaling[47]. Discovered PFS differences were of similar magnitude in both microarray-based cohorts (Fig. 6b, d) and remained significant for OS and after crossing the technological border to RNA-sequencing (Supplementary Fig. 33). This was not expected, as average survival differences based on previously classified COO-related DLBCL subtypes and underlying signatures did not generalize as well (Supplementary Fig. 31).

Our results implicate that SDCM is able to dissect overlapping biological interactions by searching for signatures of maximal GE correlation. Top genes of identified survival signatures (Fig. 5) might provide a more robust and more specific representation of genetic interactions responsible for survival differences in DLBCL than previous COO-related signatures. Signatures $k = 11$, 12, and 27 are all partially correlated, but their differences are biologically important. Similar to the 3D example (Fig. 1), partially correlated interactions do not need to be identical and should be dissected precisely.

Due to constraints such as linearity or orthogonality, previously described comparison methods have a narrower scope of detectable interactions. Hence, they often failed to dissect overlapping interaction signatures in the signal, potentially missing biologically important correlations. Our perturbative-like expansion towards monotonically non-linear interactions provides a high versatility in dissecting collectively measured molecular interactions. This might be a substantial advance and could lead to discoveries of novel correlations in transcriptome data of different cancer types with possible translational impact on diagnosis and therapy. This is a central research goal in context of the precision medicine initiative[48] that currently quantifies transcriptomes for many cancer entities. However, as the provided SDCM toolbox is based on generic mathematical concepts, namely correlation maximization, bi-monotonic regression and iterative focused dissection, it may likewise be applied to dissect many other data sources, such as mutational frequencies based on DNA-sequencing or even non-biological data such as light spectra in astrophysics[49] to infer star classes. Essentially, any data domain that was analyzed by PCA might also be suitable for dissection with

SDCM, especially if PCs were hard to interpret in the respective system context.

## Methods

**Conceptual introduction**. Here, we conceptually explain the four iterated main steps of SDCM, as introduced in the section SDCM concepts in Results.

As the number of all possible directions (GE programs) grows exponentially with data space dimension, the search strategy in step 1 concentrates on the subset of actually measured directions. Each sample column in the input matrix (each point in Fig. 1) represents one direction in the space spanned by genes. Likewise, each measured gene row defines one direction in sample space. By restricting the search to only these $n + m$ measured directions we assume that some of these candidate directions pass through point clouds that SDCM is designed to detect: We are looking for GE programs that are characterized by a consistent intrinsic order of co-expression of participating genes in multiple samples. The intrinsic gene order defines a characteristic direction in gene space. Samples expressing this program at various intensities are then scattered along that common direction, giving rise to a specific sample order. Our search functional obtains higher values for such characteristic directions than for axes that only pass through their generating data point with no others nearby. Many deviations from purely linear patterns (i.e., from constant gene co-expression ratios for all samples) are conceivable for real GE programs, as long as their characteristic co-expression order remains consistent. This has led to the model of monotonic curves. Compared to linear point clouds at orthogonal directions that can also be represented 1:1 by PCs, our class of detectable interaction signatures along nonlinear monotonic curves is huge. For this class, our above assumption for search space restriction always holds, as one participating sample is always in the middle of its directional point cloud. In contrast, SDCM does not search for, e.g., circular point clouds or points scattered in higher-dimensional manifolds of the gene space, as we do not expect such structures to exist in real GE datasets. Concentrating on the broad class of directional point clouds, our search strategy aims to select the candidate direction with the globally highest signature functional.

Such an initial direction generated by only a single data point is rarely an optimal representation of a GE program that is active in many samples. Therefore, starting from an initial direction, step 2 selects additional data points nearby to refine that direction by propagating it to the weighted mean of all selected data points (Fig. 1b). This continues iteratively, as long as the signature functional has not yet obtained its local maximum.

Once the linear direction has been optimized, step 3 regresses a curve through the data points in its vicinity. Regression weights of data points decrease with their angular distance, as illustrated by double-cones in Fig. 1c–f. Regression is guided by the key consistency constraint of our data model that after ordering gene rows and sample columns by signature strengths (given by projections on the axis from step 2) the resulting signature matrix is bimonotonic (for a detailed motivation for this requirement see the next subsection). Depicted as a heatmap after sorting by signature-specific gene and sample strengths, each regressed signature matrix $\mathbf{E}_k$ thus displays with all-monotonic rows and columns (see Fig. 2d for a high-dimensional regression result).

Real data patterns are only approximately bi-monotonic (see, e.g., Fig. 2c). Observed differences to exact bi-monotonicity might not be caused by noise alone but often contain valuable information about other overlapping GE programs. Therefore, SDCM only dissects those parts from the signal in step 4 that are exactly consistent with our bimonotonic model. The residuals remaining in the signal might then contribute to the detection of signatures dissected in later iterations.

In the following subsections, we introduce SDCM in a formal way. In particular, we detail key concepts such as the signature functional that guides the initial search of a linear axis and its optimization and our bimonotonic consistency requirements that guide signature regression onto a nonlinear monotonic curve and subsequent signal dissection.

**Formal introduction**. Here, we describe our signal and signature model again and provide an overview of the algorithmic structure, with links to detailed mathematical notes about each formal aspect.

SDCM analyzes datasets as two-dimensional matrices $\mathbf{M}_0$ with $n$ columns representing samples and $m$ rows representing genes (or other features). We aim to extract interactions as signatures defined by their signal matrices $\mathbf{E}_k$ of same size as $\mathbf{M}_0$. Components $\mathbf{E}_k(i, j) \in \mathbb{R}$ quantify the expression (or suppression) of each gene $i$ in each sample $j$ for one specific interaction. We assume that $\mathbf{M}_0$ is the result of a superposition of different $\mathbf{E}_k$, i.e., $\mathbf{M}_0 = \sum_{k=1}^{\hat{k}} \mathbf{E}_k + \beta$ where $\hat{k}$ is the initially unknown number of signatures and $\eta$ is a normal random matrix describing measurement noise. This allows genes playing multiple roles; overlapping roles may cause partial correlations between signatures that need to be dissected into individual $\mathbf{E}_k$. Similar superposition hypotheses underlie PCA, ICA, and FABIA, as these methods are equivalent to matrix factorizations of $\mathbf{M}_0$[33,37].

The dissection of $\mathbf{M}_0$ into individual $\mathbf{E}_k$ is an underdetermined problem. To overcome this, we first constrain the class of signals $\mathbf{E}_k$. We assume that biological interactions are characterized by a hierarchy of expression of participating genes ranging from weak to strong. While expression strengths may also vary between samples, we assume that the hierarchy of genes is consistent in all samples

influenced by the same interaction. Likewise, the hierarchy of samples must be consistent for all participating genes. Formally, each $\mathbf{E}_k$ is required to have a permuted gene order $\mathbf{I}_k$ and sample order $\mathbf{J}_k$ such that their resorted signal matrix $\widetilde{\mathbf{E}}_k \equiv \mathbf{E}_k(\mathbf{I}_k, \mathbf{J}_k)$ is bi-monotonic, i.e., monotonic over all genes for each sample and vice versa (see the section Model in Methods). Genes not participating in the respective interaction and unaffected samples have $\mathbf{E}_k(i, j) = 0$.

This requirement is based on the following assumption: The order of genes by their signature strengths must be consistent for each sample participating in the signature, as this gene order should represent the intrinsic structure of a single GE program that cannot change from patient to patient. Symmetrically, the order of samples by their signature strengths should be consistent for each gene participating in the signature, as this order quantifies the involvement of samples in the whole GE program that cannot change from gene to gene. Of course, a single gene of any given GE program typically cannot reliably sort samples by their involvement in the whole program due to noise or due to additional roles that gene may play in overlapping programs. However, the monotonic ties between many samples and the required repetition of that pattern in many genes and vice versa, as required by our bi-monotonic consistency model, leads to statistically significant and reliable orderings of both samples and genes. Each signature discovered by SDCM is based on such bi-monotonic ties, representing one specific underlying GE program (or another statistically significant structure in the data).

A basic example is a purely linear interaction. Here, participating genes are expressed at a fixed ratio relative to each other, inducing a gene order $\mathbf{I}$ and defining a line in gene space, such as a PC or IC (gene axis). The same interaction may differ in strength between samples, inducing a sample order $\mathbf{J}$ along that line (cf. red points in Fig. 1a). Hence, every linear interaction is automatically bi-monotonic. Bi-monotonic $\widetilde{\mathbf{E}}_k$ generalize this linear concept and are geometrically equivalent to monotonic curves (Fig. 1h). They can represent all nonlinear monotonic interactions, such as saturations (blue in Fig. 1a) or super-linear interactions (green). Furthermore, this model covers conventional clusters representing, e.g., activation thresholds (magenta). Projections of samples on such a signature curve quantify their strengths in the respective signature (see the section Signature strengths ($|\mathbf{u}^g\rangle$, $|\mathbf{u}^s\rangle$) in Methods).

In case of many gene dimensions, more complex monotonic deviations from purely linear interactions become possible. Introduced curves enable more precise approximations of such signatures, while purely linear models may lead to large residuals, effectively splitting the description of a single signature into several artificial parts. Generally, genes may interact indirectly in a complex combinatorial fashion. As GE datasets measure an ensemble of non-synchronized cells at one time point only, they usually do not provide sufficient information to extract individual gene-gene interactions. However, many well-researched gene signatures, e.g., hallmark gene sets of the Molecular Signature Database[43], show that genes realizing the same cellular function are effectively being co-expressed in the cell ensemble of affected samples. Hence, SDCM can detect signatures of such gene networks, even if participating genes are interacting only indirectly, provided the same effective co-expression is measured for several genes and samples.

Furthermore, we assume that larger signatures represent more robust interactions and that higher correlations are indicative for more specific (and less mixed) interactions. With the purpose of detecting and quantifying directions in the gene or sample space that maximize those characteristics, we defined a signature functional $\mathcal{E}$ (see the section Signature functional in Methods). Its maximization in conjunction with the signature model suffices to infer signatures from $\mathbf{M}_0$ unambiguously.

SDCM is a deterministic algorithm that detects signatures iteratively. Each iteration $k$ dissects one signature signal $\mathbf{E}_k$ from the current signal matrix $\mathbf{M}_{k-1}$ until termination, effectively dissecting $\mathbf{M}_0$ into all $\mathbf{E}_k$. Each iteration consists of four major steps:

(1) Search strategy: An initial representative (a sample or a gene) is selected that is associated with signature axes to which many other samples or genes have a high correlation (as quantified by $\mathcal{E}$). Figure 1 illustrates SDCM concepts in a 3D gene space. Figure 1a shows an initial sample and its associated gene axis. Generally, a sample or a gene may be selected as initial representative. Unlike other methods such as ICA, SDCM evaluates measured correlation structures simultaneously in both the gene and sample space to stabilize the search (see Methods/Step 1 for symmetrization details).

(2) Correlation maximization: To make the linear signature estimation robust and independent of the initial representative, we iteratively add representatives that further maximize $\mathcal{E}$ until convergence to a local maximum (Methods/Step 2). Correlation-based gene and sample weights smoothly quantify signature membership (see the section Signature focus ($|\mathbf{w}^g\rangle$, $|\mathbf{w}^s\rangle$) in Methods). Optimized signature axes represent locally maximized correlation (between signature axes and members) and maximized signature size (Fig. 1b).

(3) Bimonotonic regression: As true signatures are not necessarily linear, we employ our bimonotonic model to capture a much broader class of one-dimensional manifolds. First, the signal matrix is reordered specific to gene and sample strengths in the detected signature (see section Signature strengths ($|\mathbf{u}^g\rangle$, $|\mathbf{u}^s\rangle$) in Methods). To extract the bimonotonic signature signal, we apply regression to this sorted signal matrix (Methods/Step 3). Any residuals, i.e. parts of the signal that are not monotonic, are interpreted as

belonging to overlapping foreign signatures (or as noise). The signature signal is equivalent to a monotonic curve over the corresponding signature axis (Fig. 1c).

(4) Dissection: Finally, the detected signature signal is dissected from the overall signal matrix (Methods/Step 4). While methods based on matrix factorization or projection like PCA or ICA reduce space dimension with every PC or IC, SDCM always retains full space dimension. Rather than reducing the signal of all genes and samples by projecting the dataset along a linear axis, we utilize gene and sample weights to precisely dissect only those parts from the signal that are consistent with the detected signature (Fig. 1d). This enables dissection in cases of partially correlated signatures, i.e. when more signatures pass through the same (sub)space than this (sub)space has dimensions, e.g., four signatures in the 3D example.

These steps are repeated for every signature (Fig. 1d–g). After $\hat{k}$ dissections, the remaining signal $\mathbf{M}_{\hat{k}}$ no longer contains any qualifying signature representative and SDCM terminates (Fig. 1g). Termination determines the number of signatures in the signal via significance thresholds for correlations and signal strengths. As the sum of all $\mathbf{E}_k$ and the noise residual restores the input signal matrix $\mathbf{M}_0$, SDCM provides a complete dissection of the signal.

When dissecting high-dimensional signals, identical steps are performed for many genes in coordinate view. (For the 3D example, this coordinate view would show a heatmap with only three rows for x, y and z and as many columns as there are data points in Fig. 1.) Generally, signatures may share the same (sub)space, as already illustrated by the 3D example. Additionally, they may overlap, i.e., they may affect measurement values for the same (gene, sample)-combinations. As long as underlying correlations are sufficiently dissimilar (i.e., as long as the angle between corresponding linear signature axes is sufficiently nonzero), such overlapping signatures can still be dissected. Their regressed bimonotonic $\mathbf{E}_k$ then estimate the original non-overlapped source signals (e.g., Fig. 2d).

In the remainder of this section, we provide a comprehensive mathematical description of SDCM. For practical applications, we provide a MALTAB® analysis toolbox including the full SDCM source code, unit tests and examples (see Code availability).

**Framework**. For the mathematical description of SDCM, we use the language of vector spaces in its established notation known from quantum mechanics.

The complete measured dataset of $m$ genes (or, more generally, features) and $n$ samples (e.g. tumor samples from patients) is represented as a single matrix $\mathbf{M}_0 \in \mathbb{R}^{m \times n}$, where $\mathbb{R}^{m \times n} \equiv \{(\mathbf{X}_{ij})_{i=1\dots m, j=1\dots n} | \mathbf{X}_{ij} \in \mathbb{R}\}$ is the signal space.

The gene space $V^g$ is a vector space spanned by $m$ basis genes $\{|\mathbf{e}_i^g\rangle | i = 1\dots m\}$. Every $|\mathbf{e}_i^g\rangle$ has coordinates $(\delta_{\mu i})_{\mu=1\dots m} \in \mathbb{R}^m$ in the gene reference order $\mathbf{I}_0 \equiv (1, \dots, m)$, where $\delta_{\mu i}$ is the Kronecker delta, i.e. $\delta_{\mu i} = 1$, if $\mu = i$ and zero otherwise. We use the upper index $^g$ to indicate elements of this space. For each sample, all genes have been measured. Thus, samples are points in this vector space spanned by all basis genes. Sample $j$ is represented by the vector $|\mathbf{s}_j\rangle = \sum_{i=1}^{m} \langle \mathbf{e}_i^g | \mathbf{s}_j \rangle |\mathbf{e}_i^g\rangle$ with expression values $\langle \mathbf{e}_i^g | \mathbf{s}_j \rangle \equiv \mathbf{M}_0(i, j)$ for gene indices $i \in \{1, \dots, m\}$.

Similarly, the sample space $V^s$ is a vector space spanned by $n$ basis samples $\{|\mathbf{e}_j^s\rangle | j = 1\dots n\}$. Every $|\mathbf{e}_j^s\rangle$ has coordinates $(\delta_{\nu j})_{\nu=1\dots n} \in \mathbb{R}^n$ in the sample reference order $\mathbf{J}_0 \equiv (1, \dots, n)$. The upper index $^s$ indicates elements of this space. For each gene, all samples have been measured and hence genes are points in this vector space spanned by all basis samples. Gene $i$ is represented by the vector $|\mathbf{g}_i\rangle = \sum_{j=1}^{n} \langle \mathbf{e}_j^s | \mathbf{g}_i \rangle |\mathbf{e}_j^s\rangle$ with expression values $\langle \mathbf{e}_j^s | \mathbf{g}_i \rangle \equiv \mathbf{M}_0(i, j)$ for sample indices $j \in \{1, \dots, n\}$.

**Model**. In general terms, we aim to detect interactions in subsets of genes that exist in subsets of samples. The size of subsets is initially unknown, the participation intensity in these interactions may vary from gene to gene and from sample to sample, and subsets of different interactions may be overlapping.

We assume that the signal is a linear superposition of $\hat{k}$ signature signals $\mathbf{E}_k$ plus a noise term $\boldsymbol{\eta}$:

$$\mathbf{M}_0 = \sum_{k=1}^{\hat{k}} \mathbf{E}_k + \boldsymbol{\eta} \qquad (1)$$

This superposition model is ambiguous without further constraints. In order to maximize the range of detectable interactions, we decided to only weakly restrict the functional form of the $\mathbf{E}_k$. We require that a gene order $\mathbf{I}_k$ and a sample order $\mathbf{J}_k$ exist, such that every row and every column of the reordered matrix $\mathbf{E}_k(\mathbf{I}_k, \mathbf{J}_k)$ is monotonic. This signature model can fit any interaction following a curve in the gene space (or sample space) that is monotonic over the signature axis in the same space, as

explained in the main text. Formally, this bi-monotonicity of all $\mathbf{E}_k$ is defined by:

$$\forall k : \exists \mathbf{I}_k \in \mathrm{perm}(\mathbf{I}_0), \mathbf{J}_k \in \mathrm{perm}(\mathbf{J}_0):$$

$$\forall i \in [1, m] : \begin{pmatrix} \forall j \in [1, n-1] : \mathbf{E}_k(\mathbf{I}_k(i), \mathbf{J}_k(j+1)) \geq \mathbf{E}_k(\mathbf{I}_k(i), \mathbf{J}_k(j)) \\ \vee \forall j \in [1, n-1] : \mathbf{E}_k(\mathbf{I}_k(i), \mathbf{J}_k(j+1)) \leq \mathbf{E}_k(\mathbf{I}_k(i), \mathbf{J}_k(j)) \end{pmatrix} \quad (2)$$

$$\wedge \forall j \in [1, n] : \begin{pmatrix} \forall i \in [1, m-1] : \mathbf{E}_k(\mathbf{I}_k(i+1), \mathbf{J}_k(j)) \geq \mathbf{E}_k(\mathbf{I}_k(i), \mathbf{J}_k(j)) \\ \vee \forall i \in [1, m-1] : \mathbf{E}_k(\mathbf{I}_k(i+1), \mathbf{J}_k(j)) \leq \mathbf{E}_k(\mathbf{I}_k(i), \mathbf{J}_k(j)) \end{pmatrix},$$

where $\mathrm{perm}(\mathbf{X}) \equiv \{\pi(\mathbf{X}) | \pi : \mathbf{X} \to \mathbf{X} \text{ bijective}\}$ is the set of all permutations of a finite set $\mathbf{X}$.

To determine signatures, we utilize a maximization principle as additional constraint. As detailed in step 2 below, we iteratively maximize the signature functional $\mathcal{E}$ that quantifies both the signature size and correlations of genes and samples to signature axes based on our measure for interactions defined next.

**Measure for interactions**. To define signatures of interaction, we first have to quantify how the interactions we look for can be observed in the measured data.

We correlate genes or samples $|\mathbf{x}\rangle$ with a signature axis $|\mathbf{a}\rangle$ that linearly approximates a potential direction of interaction in the respective vector space by computing the uncentered weighted correlation defined by

$$[\mathbf{x}|\mathbf{a}]_{|\mathbf{w}\rangle} \equiv \frac{\langle \mathbf{w}.\mathbf{x}|\mathbf{w}.\mathbf{a}\rangle}{\|\mathbf{w}.\mathbf{x}\| \|\mathbf{w}.\mathbf{a}\|}, \quad (3)$$

where $|\mathbf{w}\rangle$ are context-dependent weights and dots denote component-wise products. $[\mathbf{x}|\mathbf{a}]_{|\mathbf{w}\rangle}$ measures the consistency of regulation as it only depends on directions of $|\mathbf{x}\rangle$ and $|\mathbf{a}\rangle$.

In situations where signal strengths are important, we utilize the weighted projection of $|\mathbf{x}\rangle$ on $|\mathbf{a}\rangle$

$$\langle \mathbf{x}|\mathbf{a}\rangle^0_{|\mathbf{w}\rangle} \equiv \frac{\langle \mathbf{w}.\mathbf{x}|\mathbf{w}.\mathbf{a}\rangle}{\|\mathbf{w}.\mathbf{a}\|}. \quad (4)$$

The upper index $^0$ indicates normalization by $\|\mathbf{w}.\mathbf{a}\|$.

**Algorithm overview**. SDCM is performed iteratively. In every iteration $k$, a signature is detected and its contribution to the signal is regressed, yielding the signature signal $\mathbf{E}_k$. The remaining signal

$$\mathbf{M}_k \equiv \mathbf{M}_{k-1} - \mathbf{E}_k = \mathbf{M}_0 - \sum_{l=1}^{k} \mathbf{E}_l \quad (5)$$

is the input for the next iteration. This continues until there are no more qualifying signature axes. Each detection and dissection iteration processes the following four steps.

**Step 1 Initial representative or termination**. We test if measured vectors in gene and sample space qualify as initial representative of some signature. To detect large and consistent signatures (having many highly correlated top genes and top samples) first, we search for maximal signature functional $\mathcal{E}$ by the following loop.

For each gene $|\mathbf{g}_i\rangle$ and each sample $|\mathbf{s}_j\rangle$ of the signal matrix $\mathbf{M}_{k-1}$

- Compute the following initial signature vectors:

  - For gene $|\mathbf{g}_i\rangle$ choose $|\mathbf{a}^s\rangle \equiv |\mathbf{g}_i\rangle$ as sample axis, for sample $|\mathbf{s}_j\rangle$ choose $|\mathbf{a}^g\rangle \equiv |\mathbf{s}_j\rangle$ as gene axis.
  - Compute initial weights $|\mathbf{w}^s_{\mathrm{initial}}\rangle$ respectively $|\mathbf{w}^g_{\mathrm{initial}}\rangle$ based on the standardized signal (computed as described in Supplementary Note 1).
  - Complete the pair of signature axes $(|\mathbf{a}^g\rangle, |\mathbf{a}^s\rangle)$ via weighted projections:
    - For gene $|\mathbf{g}_i\rangle$, define the gene axis $|\mathbf{a}^g\rangle$ by computing projections

$$\langle \mathbf{e}^g_{i'}|\mathbf{a}^g\rangle \equiv \langle \mathbf{g}_{i'}|\mathbf{g}_i\rangle^0_{|\mathbf{w}^s_{\mathrm{initial}}\rangle} / \|\mathbf{w}^s_{\mathrm{initial}}\| \quad (6)$$

of all other genes $|\mathbf{g}_{i'}\rangle$ onto $|\mathbf{g}_i\rangle$. Normalizing by $\|\mathbf{w}^s_{\mathrm{initial}}\|$ yields values that can directly be interpreted in units of the input signal.
    - For sample $|\mathbf{s}_j\rangle$, define the sample axis $|\mathbf{a}^s\rangle$ by computing projections

$$\langle \mathbf{e}^s_{j'}|\mathbf{a}^s\rangle \equiv \langle \mathbf{s}_{j'}|\mathbf{s}_j\rangle^0_{|\mathbf{w}^g_{\mathrm{initial}}\rangle} / \|\mathbf{w}^g_{\mathrm{initial}}\| \quad (7)$$

of all other sample $|\mathbf{s}_{j'}\rangle$ onto $|\mathbf{s}_j\rangle$.
    - The completed axes pair $(|\mathbf{a}^g\rangle, |\mathbf{a}^s\rangle)$ points to a potential signature. This pair is the basis of all subsequent computations. In this way, a symmetric treatment of genes and samples is guaranteed.
  - To quantify the relationship of all genes and samples to axes $(|\mathbf{a}^g\rangle, |\mathbf{a}^s\rangle)$, compute uncentered weighted correlations $(|\mathbf{r}^g\rangle, |\mathbf{r}^s\rangle)$ to them via

$$\langle \mathbf{e}^g_{i'}|\mathbf{r}^g\rangle \equiv [\mathbf{g}_{i'}|\mathbf{a}^s]_{|\mathbf{w}^s_{\mathrm{initial}}\rangle} \text{ and } \langle \mathbf{e}^s_{j'}|\mathbf{r}^s\rangle \equiv [\mathbf{s}_{j'}|\mathbf{a}^g]_{|\mathbf{w}^g_{\mathrm{initial}}\rangle} \quad (8)$$

- Compute probabilities $(|\mathbf{p}^g\rangle, |\mathbf{p}^s\rangle)$ that these correlations $(|\mathbf{r}^g\rangle, |\mathbf{r}^s\rangle)$ have been caused by noise (see significance of correlations in Supplementary Note 2).

- Refine signature vectors to make them more specific:

  - Define more specific weights $(|\mathbf{w}^g\rangle, |\mathbf{w}^s\rangle)$ (and $(|\mathbf{v}^g\rangle, |\mathbf{v}^s\rangle)$) based on now available correlations $(|\mathbf{r}^g\rangle, |\mathbf{r}^s\rangle)$ and their $(|\mathbf{p}^g\rangle, |\mathbf{p}^s\rangle)$ values. We refer to these weights as the (extended) signature focus, as they are used to focus subsequent computations on those genes and samples that are most correlated to the detected underlying interaction (see the section Signature focus $(|\mathbf{w}^g\rangle, |\mathbf{w}^s\rangle)$ in Methods).
  - For gene $|\mathbf{g}_i\rangle$, refine its gene axis $|\mathbf{a}^g\rangle$ by $\langle \mathbf{e}^g_{i'}|\mathbf{a}^g\rangle \equiv \langle \mathbf{g}_{i'}|\mathbf{g}_i\rangle^0_{|\mathbf{w}^s\rangle} / \|\mathbf{w}^s\|$. For sample $|\mathbf{s}_j\rangle$, refine its sample axis $|\mathbf{a}^s\rangle$ by $\langle \mathbf{e}^s_{j'}|\mathbf{a}^s\rangle \equiv \langle \mathbf{s}_{j'}|\mathbf{s}_j\rangle^0_{|\mathbf{w}^g\rangle} / \|\mathbf{w}^g\|$.
  - Refine and update correlations $(|\mathbf{r}^g\rangle, |\mathbf{r}^s\rangle)$ to signature axes: $\langle \mathbf{e}^g_{i'}|\mathbf{r}^g\rangle \equiv [\mathbf{g}_{i'}|\mathbf{a}^s]_{|\mathbf{w}^s\rangle}$ and $\langle \mathbf{e}^s_{j'}|\mathbf{r}^s\rangle \equiv [\mathbf{s}_{j'}|\mathbf{a}^g]_{|\mathbf{w}^g\rangle}$.
  - Update probabilities $(|\mathbf{p}^g\rangle, |\mathbf{p}^s\rangle)$ that observed correlations may have been caused by noise.
    - To quantify the consistency and size of a potential signature along axes $(|\mathbf{a}^g\rangle, |\mathbf{a}^s\rangle)$, compute the signature functional $\mathcal{E}[|\mathbf{a}^g\rangle, |\mathbf{a}^s\rangle]$ (see the section Signature functional in Methods).
    - To determine if axes $(|\mathbf{a}^g\rangle, |\mathbf{a}^s\rangle)$ qualify for further processing, several thresholds for signature size or significance of correlation and signal strength are tested (see Supplementary Notes 2 and 3 and default qualification thresholds in Supplementary Note 4). Default thresholds have been optimized to exclude false negative signatures as primary objective and minimize false positives as secondary objective.

If qualified axes pairs $(|\mathbf{a}^g\rangle, |\mathbf{a}^s\rangle)$ are obtained by the loop, select the pair with the highest $\mathcal{E}[|\mathbf{a}^g\rangle, |\mathbf{a}^s\rangle]$ and pass corresponding signature vectors on to step 2.

If no gene or sample in the current signal matrix $\mathbf{M}_{k-1}$ yields qualifying signature axes, SDCM terminates.

For performance optimization, we do not process every gene and sample for each iteration $k$. Instead we presort all genes and samples based on their uncentered standard deviations and apply a lookahead scheme (see presort order and lookahead scheme in Supplementary Note 5). This may slightly influence the order in which existing signatures are detected, but not whether they are detected. In any case, SDCM stays deterministic.

**Step 2 Signature axes via maximization principle**. To make identified signature axes independent of individual features of the initial representative, additional representative genes and/or samples are collected and combined until signature axes have converged (as in Fig. 1b).

Denote signature axes based on $l$ representatives by $(|\mathbf{b}^g_l\rangle, |\mathbf{b}^s_l\rangle)$. For $l = 1$, these vectors are defined by $|\mathbf{a}^g\rangle$ and $|\mathbf{a}^s\rangle$ from step 1. The same index $l$ is used for $|\mathbf{w}\rangle$, $|\mathbf{v}\rangle$, $|\mathbf{r}\rangle$ and $|\mathbf{p}\rangle$.

While signature axes have not converged or are not based on sufficiently many representatives (see convergence criteria for step 2 in Supplementary Note 6), compute the following:

- Sort all genes and/or samples in descending order by their absolute correlations to current axes $(|\mathbf{b}^g_l\rangle, |\mathbf{b}^s_l\rangle)$, i.e. by $|\langle \mathbf{e}^g_i|\mathbf{r}^g_l\rangle|$ and by $|\langle \mathbf{e}^s_j|\mathbf{r}^s_l\rangle|$. Then, compute updated signature axes for the top candidates in this order as follows and finally select the one leading to maximal signature functional.

  - Test the gene or sample as representative $l + 1$ by computing accumulated signature axes $(|\mathbf{b}^g_{l+1}\rangle, |\mathbf{b}^s_{l+1}\rangle)$ as follows:
    - For all representatives $l' = 1 \dots l+1$, denote their individual signature axes as $(|\mathbf{a}^g_{l'}\rangle, |\mathbf{a}^s_{l'}\rangle)$ (each computed as described in step 1).
    - The accumulated sample axis $|\mathbf{b}^s_{l+1}\rangle \in V^s$ is defined as arithmetic average of all individual sample axes $|\mathbf{a}^s_{l'}\rangle$, using correlations to the current sample axis $|\mathbf{b}^s_l\rangle$ as weights:

$$|\mathbf{b}^s_{l+1}\rangle \equiv \sum_{l'=1}^{l+1} [\mathbf{a}^s_{l'}|\mathbf{b}^s_l]_{|\mathbf{w}^s_l\rangle} |\mathbf{a}^s_{l'}\rangle \Big/ \sum_{l'=1}^{l+1} \left| [\mathbf{a}^s_{l'}|\mathbf{b}^s_l]_{|\mathbf{w}^s_l\rangle} \right| \quad (9)$$

  - Analogously, the accumulated gene axis $|\mathbf{b}^g_{l+1}\rangle \in V^g$ is defined as:

$$|\mathbf{b}^g_{l+1}\rangle \equiv \sum_{l'=1}^{l+1} [\mathbf{a}^g_{l'}|\mathbf{b}^g_l]_{|\mathbf{w}^g_l\rangle} |\mathbf{a}^g_{l'}\rangle \Big/ \sum_{l'=1}^{l+1} \left| [\mathbf{a}^g_{l'}|\mathbf{b}^g_l]_{|\mathbf{w}^g_l\rangle} \right| \quad (10)$$

  - As in step 1, compute correlations $(|\mathbf{r}^g_{l+1}\rangle, |\mathbf{r}^s_{l+1}\rangle)$ to the accumulated axes, i.e.

$$\langle \mathbf{e}^g_{i'}|\mathbf{r}^g_{l+1}\rangle \equiv [\mathbf{g}_{i'}|\mathbf{b}^s_{l+1}]_{|\mathbf{w}^s_l\rangle} \text{ and } \langle \mathbf{e}^s_{j'}|\mathbf{r}^s_{l+1}\rangle \equiv [\mathbf{s}_{j'}|\mathbf{b}^g_{l+1}]_{|\mathbf{w}^g_l\rangle} \quad (11)$$

- Compute their $p$-values $(|\mathbf{p}_{l+1}^{g}\rangle, |\mathbf{p}_{l+1}^{s}\rangle)$ as well.
- Compute the signature focus $(|\mathbf{w}_{l+1}^{g}\rangle, |\mathbf{w}_{l+1}^{s}\rangle)$ and the extended signature focus $(|\mathbf{v}_{l+1}^{g}\rangle, |\mathbf{v}_{l+1}^{s}\rangle)$ for these correlations as in step 1.
- Compute the signature functional $\mathcal{E}[|\mathbf{b}_{l+1}^{g}\rangle, |\mathbf{b}_{l+1}^{s}\rangle]$ and store it for the tested gene or sample.

- Finally, select that gene or sample as representative $l + 1$ that leads to the highest signature functional.

Once convergence criteria have been met for a certain $\hat{l}$, $\left(\left|\mathbf{b}_{\hat{l}}^{g}\right\rangle, \left|\mathbf{b}_{\hat{l}}^{s}\right\rangle\right)$ are the final signature axes. They linearly approximate contributions of the detected interaction to the signal, similar to PCs or ICs. In step 3, they serve as basis for a monotonic nonlinear (and hence more precise) approximation.

**Step 3 Bi-monotonic regression and smoothening.** The contributions of a signature to the signal matrix $\mathbf{M}_{k-1}$ are regressed by one outer and one inner loop until convergence. The outer loop (with index $o$) is structured as follows:

- Compute signature strengths $|\mathbf{u}_{k,o}^{g}\rangle$ for genes and $|\mathbf{u}_{k,o}^{s}\rangle$ for samples (e.g., overlaid curves in Fig. 2c; see the section Signature strengths $(|\mathbf{u}^{g}\rangle, |\mathbf{u}^{s}\rangle)$ in Methods).
- Determine signature orders $(\mathbf{I}_{k,o}, \mathbf{J}_{k,o})$ as follows. The gene order $\mathbf{I}_{k,o}$ sorts genes by $\langle \mathbf{e}_{i}^{g} | \mathbf{u}_{k,o}^{g} \rangle$, i.e.,

$$\forall i \in [1, m-1] : \left\langle \mathbf{e}_{\mathbf{I}_{k,o}(i+1)}^{g} \middle| \mathbf{u}_{k,o}^{g} \right\rangle \geq \left\langle \mathbf{e}_{\mathbf{I}_{k,o}(i)}^{g} \middle| \mathbf{u}_{k,o}^{g} \right\rangle \tag{12}$$

- Likewise, the sample order $\mathbf{J}_{k,o}$ sorts samples by $\langle \mathbf{e}_{j}^{s} | \mathbf{u}_{k,o}^{s} \rangle$, i.e.,

$$\forall j \in [1, n-1] : \left\langle \mathbf{e}_{\mathbf{J}_{k,o}(j+1)}^{s} \middle| \mathbf{u}_{k,o}^{s} \right\rangle \geq \left\langle \mathbf{e}_{\mathbf{J}_{k,o}(j)}^{s} \middle| \mathbf{u}_{k,o}^{s} \right\rangle \tag{13}$$

- Resort rows and columns of the current signal matrix $\mathbf{M}_{k-1}$ by these signature orders. This yields an already roughly bi-monotonic signal $\mathbf{M}_{k-1}(\mathbf{I}_{k,o}, \mathbf{J}_{k,o})$ (e.g., the heatmap in Fig. 2c). However, this sorted signal matrix still needs to be regressed and smoothened to obtain exact bi-monotonicity and finally to subtract solely the signature's contributions to the signal. (Directly subtracting the sorted but otherwise unmodified $\mathbf{M}_{k-1}(\mathbf{I}_{k,o}, \mathbf{J}_{k,o})$ would lead to a zero signal, i.e., a loss of all information.)
- Compute the bi-monotonically regressed matrix $\widetilde{\mathbf{M}}_{o,\hat{q}}$ for the presorted signal matrix $\mathbf{M}_{k-1}(\mathbf{I}_{k,o}, \mathbf{J}_{k,o})$ by processing the inner convergence loop with index $q$, where $\hat{q}$ denotes the inner iteration that reached convergence (see the section Bi-monotonic regression in Methods).
- Apply adaptive smoothening $\mathcal{S}_{|\mathbf{u}_{k,o}^{g}\rangle, |\mathbf{u}_{k,o}^{s}\rangle}(\widetilde{\mathbf{M}}_{o,\hat{q}})$ by rescaling to the signature strengths space and via 2D Fourier transforms (see the section Smoothening operator in Methods). The smoothened matrix is still bi-monotonic, as smoothening is a local operation that cannot change monotonicity.
- Once convergence criteria are met for a certain $\hat{o}$ (Supplementary Note 7), $|\mathbf{u}_{k,\hat{o}}^{g}\rangle$ and $|\mathbf{u}_{k,\hat{o}}^{s}\rangle$ are the final signature strengths and the result $\mathcal{S}_{|\mathbf{u}_{k,\hat{o}}^{g}\rangle, |\mathbf{u}_{k,\hat{o}}^{s}\rangle}(\widetilde{\mathbf{M}}_{\hat{o},\hat{q}})$ is passed on to step 4.

**Step 4 Signature signal $\mathbf{E}_{k}$ and its dissection.** Dissection strengths $\mathbf{D}_{k} \in \mathbb{R}^{m \times n}$ for signature $k$ are defined as square roots of the components of the outer weights product $|\mathbf{w}_{\hat{l}}^{g}\rangle \otimes |\mathbf{w}_{\hat{l}}^{s}\rangle$, i.e.,

$$\mathbf{D}_{k}(i, j) = \sqrt{\langle \mathbf{e}_{i}^{g} | \mathbf{w}_{\hat{l}}^{g} \rangle \langle \mathbf{e}_{j}^{s} | \mathbf{w}_{\hat{l}}^{s} \rangle} \tag{14}$$

$\widetilde{\mathbf{D}} \equiv \mathbf{D}(\mathbf{I}_{k,\hat{o}}, \mathbf{J}_{k,\hat{o}})$ denotes these dissection strengths in the final signature order. The signature signal is defined in signature order as component-wise product

$$\widetilde{\mathbf{E}}_{k} \equiv \widetilde{\mathbf{D}}.\mathcal{S}_{|\mathbf{u}_{k,\hat{o}}^{g}\rangle, |\mathbf{u}_{k,\hat{o}}^{s}\rangle}(\widetilde{\mathbf{M}}_{\hat{o},\hat{q}}) \tag{15}$$

(e.g., Fig. 2d; gray shadings depict $\widetilde{\mathbf{D}}$ and fully grayed-out pixels correspond to zero dissection strengths).

The signature signal in reference order is obtained by sorting back via $\mathbf{E}_{k}(\mathbf{I}_{k,\hat{o}}, \mathbf{J}_{k,\hat{o}}) \equiv \widetilde{\mathbf{E}}_{k}$.

Dissection of signature $k$ is now simply a matrix subtraction of its signature signal $\mathbf{E}_{k}$ from the current signal $\mathbf{M}_{k-1}$. The remaining signal

$$\mathbf{M}_{k} = \mathbf{M}_{k-1} - \mathbf{E}_{k} = \mathbf{M}_{0} - \sum_{l=1}^{k} \mathbf{E}_{l} \tag{16}$$

is the input for the next detection and dissection iteration $k + 1$ (e.g., Fig. 2e or Fig. 1d).

**Signature focus.** Purpose of the signature focus is to define where a detected effect ends. While this may seem easy for plateau-like clusters (having few highly correlated genes or samples while all others only have low correlations), often real effects have no clear edge. In this case, we prefer a smooth decrease in membership weights. The signature focus should not influence ranks of top members, as they need to be determined by signature strengths for optimal bimonotonic regression (cf. Eqs. 29 and 30). But non-members should be identified and excluded by the signature focus to minimize the influence of noise.

The signature focus consists of correlation-based gene and sample weights that allow the computation of all vectors and scores as specific as possible, even if the detected interaction only affects a small subset of measured genes and samples. To retain specificity for such small signatures, we set weight components $\langle \mathbf{e}_{i}^{g} | \mathbf{w}^{g} \rangle$ and $\langle \mathbf{e}_{j}^{s} | \mathbf{w}^{s} \rangle$ exactly to zero for all (and potentially very many) genes and samples that have only weak or insignificant correlations to detected signature axes.

For samples $|\mathbf{e}_{j}^{s}\rangle$, let

$$x_{j}^{s} \equiv \left| \left\langle \mathbf{e}_{j}^{s} \middle| \mathbf{r}^{s} \right\rangle \right| \cdot \left( 1 - \left\langle \mathbf{e}_{j}^{s} \middle| \mathbf{p}^{s} \right\rangle \right)^{2}, \tag{17}$$

where the second factor decreases quadratically to zero with the noise probability of sample correlations (cf. Supplementary Note 2). Sample weights are defined relative to $x_{j}^{s}$. Values ≥50% of the maximum of all $x_{j}^{s}$ are already mapped to full weight:

$$y_{j}^{s} \equiv \min\left( 1, x_{j}^{s} \middle/ \left( \frac{1}{2} \max_{j'}\left( x_{j'}^{s} \right) \right) \right) \tag{18}$$

Consequently, weights have no influence on the signature's order of top samples, as is intended. To exclude any unspecific influence of samples with relatively weak or insignificant correlation, we finally set all weights to zero that are lower than two thirds of their quantile:

$$\langle \mathbf{e}_{j}^{s} | \mathbf{w}^{s} \rangle \equiv \begin{cases} y_{j}^{s}, & \text{if } y_{j}^{s} \geq \frac{2}{3n} \left| \left\{ y_{j'}^{s} \middle| y_{j'}^{s} \leq y_{j}^{s} \right\} \right| \\ & \text{and } 0 \text{ otherwise} \end{cases} \tag{19}$$

Analogously, gene weights $|\mathbf{w}^{g}\rangle$ are given by

$$\langle \mathbf{e}_{i}^{g} | \mathbf{w}^{g} \rangle \equiv \begin{cases} y_{i}^{g}, & \text{if } y_{i}^{g} \geq \frac{2}{3m} \left| \left\{ y_{i'}^{g} \middle| y_{i'}^{g} \leq y_{i}^{g} \right\} \right| \\ & \text{and } 0 \text{ otherwise} \end{cases} \tag{20}$$

where $y_{i}^{g} \equiv \min\left( 1, x_{i}^{g} \middle/ \left( \frac{1}{2} \max_{i'}(x_{i'}^{g}) \right) \right)$ and $x_{i}^{g} \equiv |\langle \mathbf{e}_{i}^{g} | \mathbf{r}^{g} \rangle| \cdot (1 - \langle \mathbf{e}_{i}^{g} | \mathbf{p}^{g} \rangle)^{2}$.

For signature size estimation and qualification thresholds, mapping all $x$-values above 50% to full weight is not optimal. To keep the full dynamic range of weights for these tasks, we additionally define the extended signature focus

$$(|\mathbf{v}^{g}\rangle, |\mathbf{v}^{s}\rangle) \tag{21}$$

by increasing the upper threshold from 50% to 100% (i.e., $y_{i}^{g} \equiv x_{i}^{g}$ and $y_{j}^{s} \equiv x_{j}^{s}$) and by decreasing the lower specificity threshold from two thirds to 0.4. Otherwise, weights $(|\mathbf{v}^{g}\rangle, |\mathbf{v}^{s}\rangle)$ and $(|\mathbf{w}^{g}\rangle, |\mathbf{w}^{s}\rangle)$ are computed in the same way.

Importantly, specificity thresholds of the signature focus exclude noise genes from computation that could otherwise reduce both specificity of signatures and detection robustness. In case of many measured genes and small signatures, this can also speed up computation.

**Signature functional.** To detect and unambiguously dissect a signature, first its optimal linear directions along which it extends in gene and sample space have to be determined. The goal of the signature functional is to score candidate directions during the initial search (Methods/Step 1) and during optimization (Methods/Step 2).

The signature functional $\mathcal{E}[|\mathbf{a}^{g}\rangle, |\mathbf{a}^{s}\rangle]$ assigns to every possible axes pair $(|\mathbf{a}^{g}\rangle, |\mathbf{a}^{s}\rangle)$ a scalar $\in \mathbb{R}$. $\mathcal{E}$ is the larger, the more genes or samples are correlated to these axes, and the higher these correlations are. The selection of an initial representative with help of this functional (step 1) and maximizing this functional (step 2) guides SDCM to one of the largest and most consistent signatures in the current signal $\mathbf{M}_{k-1}$. (Thus, in all versatility tests, the large signature #1 was always detected first, and narrow or weak signatures #5, #6, and #7 were detected last; see detection ranks in Supplementary Fig. 1.)

The signature size, i.e., the number of participating genes or samples, is estimated by summing gene weights $v_{i}^{g} \equiv \langle \mathbf{e}_{i}^{g} | \mathbf{v}^{g} \rangle$ respectively sample weights $v_{j}^{s} \equiv \langle \mathbf{e}_{j}^{s} | \mathbf{v}^{s} \rangle$ of the extended signature focus:

$$m_{k} \equiv \sum_{i=1}^{m} v_{i}^{g} \quad \text{and} \quad n_{k} \equiv \sum_{j=1}^{n} v_{j}^{s} \tag{22}$$

Average absolute correlations for all genes and all samples in the extended signature focus are computed as weighted means:

$$r_{k}^{g} \equiv \left( \sum_{i=1}^{m} v_{i}^{g} \middle| r_{i}^{g} \middle| \right) \middle/ m_{k} \quad \text{and} \quad r_{k}^{s} \equiv \left( \sum_{j=1}^{n} v_{j}^{s} \middle| r_{j}^{s} \middle| \right) \middle/ n_{k} \tag{23}$$

Next, these separate scores for genes and samples need to be combined. As $m_{k} n_{k}$ corresponds to the number of measurement values supporting the signature, a natural choice for the combined signature size would be their geometric average $\sqrt{m_{k} n_{k}}$. However, signatures affecting the same number $m_{k} n_{k}$ of measured (gene, sample)-values in $\mathbf{M}_{0}$ may be caused by noise or by artifacts with

different probabilities. For example, a narrow signature concerning 10,000 genes in two samples is more likely a measurement artefact than a signature representing an interaction that affects 100 genes in 200 samples. To detect and dissect broad and robust signatures first, we therefore optimized the usual geometric average by putting 90% weight on the order dimension with lower size. Hence, the combined signature size is defined as

$$s_k \equiv \sqrt{\min(m_k, n_k)^{0.9} \max(m_k, n_k)^{0.1}} \tag{24}$$

Putting 90% weight on $\min(m_k, n_k)$ effectively increases the signature functional for the broad signature of same area and decreases it for narrow stripe-like signatures. The latter will still be detected, but in a later detection iteration.

A natural choice to combine the average correlations $r_k^g$ for genes and $r_k^s$ for samples is an arithmetic average that takes into account the number of measured values underlying the computation of these weighted correlations:

$$r_k \equiv \left(n_k r_k^g + m_k r_k^s\right)/(n_k + m_k) \tag{25}$$

To put emphasis on the most consistent signatures (that have average correlations of $r_k$ near one) and to detect them first, we modify this to $r_k/\sqrt{1 - r_k^2}$ (inspired by the $t$-statistic of a correlation; cf. significance of correlations in Supplementary Note 2).

Hence, our final signature functional is defined by:

$$\mathcal{E}[|\mathbf{a}^g\rangle, |\mathbf{a}^s\rangle] \equiv s_k r_k / \sqrt{1 - r_k^2} \tag{26}$$

**Signature strengths**. As preparation for bi-monotonic regression, genes and samples need to be sorted such that their measured signals in the signature focus are arranged as bimonotonic as possible with respect to detected signature axes. To achieve this, we sort them by their signature strengths. Whereas correlations only quantify the consistency of observed GE data with a signature axis, signature strengths are additionally proportional to the magnitude of observed signals in direction of a signature axis. In this way, signature strengths reflect gene activities in the underlying interaction (inducing a gene order) or, respectively, the activities of this interaction in samples (inducing a sample order). Signature strengths are also useful candidates for downstream analyses, e.g., they may serve as covariates in Cox survival models.

For the first regression iteration $o = 1$ (in step 3), the gene strengths vector $|\mathbf{u}_{k,1}^g\rangle \in V^g$ is defined by

$$\left\langle \mathbf{e}_i^g | \mathbf{u}_{k,1}^g \right\rangle \equiv \left\langle \mathbf{g}_i | \mathbf{b}_l^s \right\rangle_{\big|\mathbf{w}_l^s}^0 / \left\| \mathbf{w}_l^s \right\|, \tag{27}$$

i.e. by weighted projections of each gene on the detected sample axis $|\mathbf{b}_l^s\rangle$ in the final sample focus $|\mathbf{w}_l^s\rangle$, normalized by $\|\mathbf{w}_l^s\|$. This normalization guarantees values that can be directly interpreted in units of measurement values (i.e., typically as $\log_2$(ratios) for GE data). Likewise, the sample strength vector $|\mathbf{u}_{k,1}^s\rangle \in V^s$ for $o = 1$ is defined by

$$\left\langle \mathbf{e}_j^s | \mathbf{u}_{k,1}^s \right\rangle \equiv \left\langle \mathbf{s}_j | \mathbf{b}_l^g \right\rangle_{\big|\mathbf{w}_l^g}^0 / \left\| \mathbf{w}_l^g \right\|. \tag{28}$$

For higher precision, we utilize regressed signature curves from the last iteration $o - 1$ as projection targets instead of signature axes for all iterations $o > 1$. Signature curves are defined as projections of the bi-monotonically regressed signal matrix $\widetilde{\mathbf{X}}$. The associated gene curve is a vector-valued function $|\mathbf{c}^g\rangle$ of sample indices $j$ defined by $\langle \mathbf{e}_i^g | \mathbf{c}^g(j) \rangle \equiv \widetilde{\mathbf{X}}(i, j)$. Analogously, the associated sample curve $|\mathbf{c}^s\rangle$ is a vector-valued function of gene indices $i$ defined by $\langle \mathbf{e}_j^s | \mathbf{c}^s(i) \rangle \equiv \widetilde{\mathbf{X}}(i, j)$. For any iteration $o > 1$, we consider signature curves for the regression result from the last iteration $o - 1$, i.e., for $\widetilde{\mathbf{X}} \equiv \mathcal{S}_{|\mathbf{u}_{k,o-1}^g\rangle, |\mathbf{u}_{k,o-1}^s\rangle}(\widetilde{\mathbf{M}}_{o-1,\hat{q}})$ (see algorithm step 3). With these signature curves as targets, again weighted projections are utilized to determine refined signature strengths:

$$\left\langle \mathbf{e}_i^g | \mathbf{u}_{k,o}^g \right\rangle \equiv \left\langle \mathbf{g}_i | \mathbf{c}_{k,o-1}^s(i) \right\rangle_{\big|\mathbf{w}_l^s}^0 / \left\| \mathbf{w}_l^s \right\| \tag{29}$$

$$\left\langle \mathbf{e}_j^s | \mathbf{u}_{k,o}^s \right\rangle \equiv \left\langle \mathbf{s}_j | \mathbf{c}_{k,o-1}^g(j) \right\rangle_{\big|\mathbf{w}_l^g}^0 / \left\| \mathbf{w}_l^g \right\| \tag{30}$$

**Smoothening operator**. Purpose of smoothening is that the signature signal becomes (nearly) the same for all genes and samples with (nearly) equal signature strengths. In particular, the signature signal for all genes with signature strengths near zero, i.e., all genes outside of the signature's focus, are averaged by smoothening. Other than regression, smoothening is a local operation that cannot guarantee monotonicity if applied to a non-monotonic signal. However, if applied to an already monotonic signal, this monotonicity is always preserved.

Let $\mathbf{I}$ denote the gene order by components of gene strengths $|\mathbf{u}^g\rangle$, $\mathbf{J}$ the sample order by components of sample strengths $|\mathbf{u}^s\rangle$ and $\widetilde{\mathbf{X}} \equiv \mathbf{X}(\mathbf{I}, \mathbf{J})$ a sorted signal matrix. To compute the adaptive smoothening $\mathcal{S}_{|\mathbf{u}^g\rangle, |\mathbf{u}^s\rangle}(\widetilde{\mathbf{X}})$, $\widetilde{\mathbf{X}}$ is first rescaled using signature strengths $|\mathbf{u}^g\rangle$ and $|\mathbf{u}^s\rangle$. A resolution of $m^- \equiv 512$ rows and $n^- \equiv 512$ columns is sufficient for this rescaled space, as only monotonic changes

need to be resolved. Basis vectors $|\mathbf{e}_{i^-}^{g-}\rangle$ of the rescaled (and downscaled) gene space correspond for $i^- \in [1, m^-]$ to equidistant gene strengths $\mathbf{u}_{\mathbf{I}(1)}^g + (-\mathbf{u}_{\mathbf{I}(1)}^g + \mathbf{u}_{\mathbf{I}(m)}^g) \cdot \frac{i^-}{m^-}$, where $\mathbf{u}_{\mathbf{I}(i)}^g$ abbreviates $\langle \mathbf{e}_i^g | \mathbf{u}^g \rangle$. Likewise, basis vectors $|\mathbf{e}_{j^-}^{s-}\rangle$ of the rescaled and downscaled sample space correspond for $j^- \in [1, n^-]$ to equidistant sample strengths $\mathbf{u}_{\mathbf{J}(1)}^s + (-\mathbf{u}_{\mathbf{J}(1)}^s + \mathbf{u}_{\mathbf{J}(n)}^s) \cdot \frac{j^-}{n^-}$. The rescaled and downscaled signal $\widetilde{\mathbf{X}}^- \in \mathbb{R}^{m^- \times n^-}$ is computed by arithmetic averaging of $\widetilde{\mathbf{X}}$ components in corresponding grid cells of signature strength. If no (gene, sample) pair has signature strengths in corresponding intervals, nearest-neighbor interpolation is employed.

A smoothening window of constant width in this rescaled space of equidistant signature strengths corresponds to an adaptive smoothening window in the original signal space $\mathbb{R}^{m \times n}$. Let $\mathbf{G}_{\sigma^g, \sigma^s} \in \mathbb{R}^{m^- \times n^-}$ denote a Gaussian kernel centered at indices $(\frac{m^-}{2}, \frac{n^-}{2})$. Default standard deviations $\sigma^g$ and $\sigma^s$ correspond to eight pixels in the downscaled space $\mathbb{R}^{m^- \times n^-}$, i.e., to a smoothening window size of $\frac{1}{64}$ of the distance between minimum and maximum signature strength.

The smoothening is given by the convolution $\widetilde{\mathbf{X}}^- * \mathbf{G}_{\sigma^g, \sigma^s}$. As the direct implementation of convolutions has a complexity that is quadratic in the number of points, we apply the convolution theorem[50]:

$$\widetilde{\mathbf{X}}^- * \mathbf{G}_{\sigma^g, \sigma^s} = \mathcal{F}^{-1}\left(\mathcal{F}\left(\widetilde{\mathbf{X}}^-\right).\mathcal{F}\left(\mathbf{G}_{\sigma^g, \sigma^s}\right)\right) \tag{31}$$

This reduces the problem to the component-wise multiplication of two Fourier-transformed matrices and one inverse Fourier transform of the result. Fast Fourier transform implementations for $\mathcal{F}$ and $\mathcal{F}^{-1}$ (provided by fft2 and ifft2 functions in MATLAB®) are employed and have only log-linear complexity.

To finally obtain the smoothened signal for original gene and sample strengths, we look up values in this rescaled smoothened matrix by linear interpolation. Let $\mathcal{I}_{|\mathbf{u}^g\rangle, |\mathbf{u}^s\rangle}$ denote this linear interpolation; then the final smoothened matrix is

$$\mathcal{S}_{|\mathbf{u}^g\rangle, |\mathbf{u}^s\rangle}\left(\widetilde{\mathbf{X}}\right) \equiv \mathcal{I}_{|\mathbf{u}^g\rangle, |\mathbf{u}^s\rangle}\left(\widetilde{\mathbf{X}}^- * \mathbf{G}_{\sigma^g, \sigma^s}\right) \tag{32}$$

This signal smoothening procedure is the development result of maximizing robustness. It is robust against too strong smoothening that would otherwise introduce dissection artefacts into the signal when the signature signal changes abruptly. Additionally, it is robust against too weak smoothening which would cause a too fine-grained dissection, thereby dissecting and losing information about overlapping signatures. Both robustness goals were achieved by rescaling the signal to and smoothening it over a grid representing equidistant signature strengths. Hence, if signature strength decreases rapidly from one gene to its neighbor in signature strength order, there are many window sizes in-between (sharp signal edges are retained). Conversely, all member genes with approximately equal signature strength end up in the same smoothening window (signal roughness due to noise is completely averaged here). This adaptiveness makes smoothening robust and generic. For the same reason, our smoothening concept is largely independent of actual window width, within a certain range. The actually chosen grid size of 512*512 pixels is more than sufficient to resolve bi-monotonic patterns. As robustness was already taken care of, we optimized this resolution and the window width subsequently for 2D FFT performance (hence the powers of 2).

**Bi-monotonic regression**. To approximate interactions by our signature model as precisely as possible, we start from measured data in the signature focus ordered by signature strengths and regress it to the nearest bimonotonic representation. Residuals that do not fit bi-monotonicity are assumed to originate from overlapping distinct signatures that will be dissected in later iterations.

Bi-monotonic regression is based on weighted 1D isotonic regressions as implemented by the Generalized Pool Adjacent Violator (GPAV) algorithm[51]. The following loop until convergence (with index j) realizes the 2D bi-monotonic regression.

For $q = 0$, start with the presorted but not yet regressed signal matrix

$$\widetilde{\mathbf{M}}_{o,0} \equiv \mathbf{M}_{k-1}\left(\mathbf{I}_{k,o}, \mathbf{J}_{k,o}\right) \tag{33}$$

from the outer loop $o$ of algorithm step 3. Regression weights are initialized with

$$\mathbf{W}_0^{\widetilde{2D}} \equiv \mathbf{W}_0^{2D}\left(\mathbf{I}_{k,o}, \mathbf{J}_{k,o}\right), \tag{34}$$

where $\mathbf{W}_0^{2D} \in \mathbb{R}^{m \times n}$ is defined by the components of the outer product $|\mathbf{w}_l^g\rangle \otimes |\mathbf{w}_l^s\rangle$.

Each iteration $q$ of the convergence loop is structured as follows:

- 1D regression of genes: Apply GPAV to every gene row in $\widetilde{\mathbf{M}}_{o,q}$ and use corresponding rows of $\mathbf{W}_q^{\widetilde{2D}}$ as regression weights. This results in monotonically regressed genes that are collected again as matrix $\widetilde{\mathbf{G}}_{q+1} \in \mathbb{R}^{m \times n}$. Each gene row in $\widetilde{\mathbf{G}}_{q+1}$ is a step function that consists of blocks with constant regressed GE values, while expression values of neighboring blocks are either all monotonically increasing or all monotonically decreasing. Each block corresponds to a sample interval in the sample order $\mathbf{J}_{k,o}$. GPAV also updates sample weights (individually for each gene) by averaging input weights for each block; they are collected as rows of the matrix $\mathbf{W}_{q+1}^{\mathbf{G}}$.

- 1D regression of samples: Likewise, regressions of sample columns in $\widetilde{\mathbf{M}}_{o,q}$ are realized with GPAV, using corresponding columns of $\mathbf{W}_q^{\widetilde{2D}}$ as regression weights. This results in a matrix of monotonic columns $\widetilde{\mathbf{S}}_{q+1} \in \mathbb{R}^{m \times n}$ and in updated gene weights for each sample $\mathbf{W}_{q+1}^{\widetilde{\mathbf{S}}} \in \mathbb{R}^{m \times n}$.

- Adaptively smoothen the signal, i.e. compute $\mathcal{S}_{|\mathbf{u}_{k,o}^g\rangle,|\mathbf{u}_{k,o}^s\rangle}(\widetilde{\mathbf{M}}_{o,q})$ (see Eq. 32). This smoothening result will only be relevant below in Eq. (36) for genes and samples with weak or zero regression weights, i.e. for those outside of the signature focus.

- For a smooth and quick convergence, we combine weights symmetrically by

$$\mathbf{W}_{q+1}^{\widetilde{2D}} \equiv \left(n\mathbf{W}_q^{\widetilde{2D}} + m\mathbf{W}_{q+1}^{\widetilde{\mathbf{S}}}\right)/(2(n+m)) + \left(m\mathbf{W}_q^{\widetilde{2D}} + n\mathbf{W}_{q+1}^{\widetilde{\mathbf{G}}}\right)/(2(n+m)) \tag{35}$$

- The approximation of the bi-monotonic signal by iteration $o$ is computed as the weighted average of monotonic genes, monotonic samples and the smoothened signal for pixels with low weight:

$$\widetilde{\mathbf{M}}_{o,q+1} = \frac{\left(n\mathbf{W}_q^{\widetilde{2D}} + m\mathbf{W}_{q+1}^{\widetilde{\mathbf{S}}}\right)}{2(n+m)} \cdot \widetilde{\mathbf{G}}_{q+1} + \frac{\left(m\mathbf{W}_q^{\widetilde{2D}} + n\mathbf{W}_{q+1}^{\widetilde{\mathbf{G}}}\right)}{2(n+m)} \cdot \widetilde{\mathbf{S}}_{q+1} + \left(1 - \mathbf{W}_{q+1}^{\widetilde{2D}}\right) \cdot \mathcal{S}_{|\mathbf{u}_{k,o}^g\rangle,|\mathbf{u}_{k,o}^s\rangle}\left(\widetilde{\mathbf{M}}_{o,q}\right) \tag{36}$$

- Finally, we test for convergence. (In the default configuration, the pixel standard deviation of remaining differences $\widetilde{\mathbf{M}}_{o,q+1} - \widetilde{\mathbf{M}}_{o,q}$ must be less than $\frac{1}{1000}$ of the data noise level.) Once converged (iteration $\hat{q}$), return the bi-monotonic $\widetilde{\mathbf{M}}_{o,\hat{q}}$ to algorithm step 3.

**Algorithmic complexity.** Algorithms having a complexity that grows like the volume of a vector space, i.e., exponentially in gene or sample dimensions, are typically of no practical use when analyzing high-dimensional datasets (curse of dimensionality). Here we show that SDCM's complexity is bounded by a low-order polynomial.

Our search strategy tests up to $m + n$ initial representatives (genes or samples) to identify the one with maximal signature functional. Each candidate leads to a pair of signature axes (see the section Algorithm overview in Methods). Correlating the gene axis with all $n$ samples has complexity $O(nm)$, as each correlation is computed with linear complexity in the number of genes per sample, i.e., $O(m)$. Correlating the sample axis with all $m$ genes likewise results in $O(nm)$. Taken together, the search strategy has complexity $O((m + n) \cdot (nm + mn)) = O((m + n) \cdot nm)$. As all other steps are essentially bounded by constants (e.g., by the unknown but finite size of the signature focus), they are irrelevant for complexity estimation. In addition to input size, complexity of input signals influences runtime. Due to the iterative structure of SDCM, this translates into a linear dependency on the number of detectable signatures $\hat{k}$. Overall, SDCM's complexity is

$$O\left(\hat{k} \cdot (m + n) \cdot nm\right) \tag{37}$$

To test this complexity analysis and to quantify SDCM performance in practice, we analyzed signals for gene spaces with up to $m = 80{,}000$ dimensions. Consistently, the empirical complexity showed runtime asymptotics that are quadratic in $m$ (Supplementary Fig. 14).

**Further methods.** While not essential for core detection concepts, SDCM sub-routines are detailed in Supplementary Notes 1–7 for completeness. Procedures for method comparison and validation including details about analyzed GE datasets are provided in Supplementary Notes 8–12. Finally, patient survival analyses are detailed in Supplementary Notes 14–16.

**Reporting summary.** Further information on research design is available in the Nature Research Reporting Summary linked to this article.

## Data availability

Analyzed DLBCL GE datasets are publicly available in NCBI GEO[52] via accessions GSE31312[3] for the detection cohort and GSE10846[2] for the validation cohort. Data for the RNA-sequencing based add-on validation cohort are available from the European Genome-Phenome Archive (https://www.ega-archive.org) via accession EGAS00001002606[39]. Discovered DLBCL signatures and derived top gene sets are provided as Supplementary Data 1–3.

## Code availability

A MATLAB® analysis toolbox including the full SDCM source code is provided at https://github.com/GrauLab/SDCM (the first release commit is the paper version). The toolbox is ready to run and includes low and high-dimensional examples for unit testing and for a quick start with your own datasets (simply open selfTest_lowDim.m or selfTest_highDim.m in your MATLAB® editor and press F5 to run it). It is free to use for any academic purpose. For commercial contexts, SDCM is free for all testing and development purposes (contact otherwise). We used MATLAB® (versions R2013b-R2016b, The MathWorks® Inc., Natick, Massachusetts, United States) for development and all computations.

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

## Acknowledgements

We thank Eyke Hüllermeier for helpful discussions and comments on the manuscript. This work was supported by the Marburg Center for Synthetic Microbiology (SYN-MIKRO), by a Doctoral scholarship to M.G. from the Philipps University of Marburg as well as by research grants to G.L. from the German Research Foundation (DFG), the Deutsche Krebshilfe, the Swiss National Science Foundation, the Else Kröner-Fresenius-Stiftung, and the DFG Cells-in-Motion Cluster of Excellence (EXC 1003).

## Author contributions

P.L. coordinated research. M.G. conceived and implemented SDCM. All authors contributed to study and validation design. M.G. performed computations. All authors analyzed signatures. G.L. interpreted DLBCL biology. P.L. and M.G. formulated the algorithm mathematically. All authors wrote the manuscript.

## Competing interests

The authors declare no competing interests.
