## [Peer Review File · Nature Communications]

Reviewers' comments:

Reviewer #1 (Remarks to the Author):

Grau et al described a novel unsupervised learning concept known as Signal Dissection by Correlation Maximization (SDCM) that dissects large gene expression datasets into signatures. The authors claimed that the derived signature captures a particular signal pattern that was consistently observed in multiple genes and samples, with potential associated with the same underlying genetic interactions. The authors compared their method with other existing clustering methods, and showed that in their method could derived more predictive signatures than other methods in a malignant lymphomas data set. Using another validation set, the authors claimed that their method could derived gene expression signatures that correlated with Progression Free Survival (PFS) of malignant lymphoma in the training set, and similarly showed the same PFS significant in the validation set. Overall, the manuscript is hard to follow as it is very dense and a lot of references to supplementary results. This reviewer has the following comments:

Major comments

- 1) Unsupervised learning approach is an approach to extract novel gene-gene interactions from high-throughput omics data. However, one of the limitations of the unsupervised approach is the determination of the number of k clusters in a particular data set. The authors have showed that the number of k for a the same dataset (Fig 5), could be extracted with k=27 and k=11. However, in a realistic application, determining the best (or optimal) number of k is still unclear.
- 2) Figure 3 is very hard to interpret, as it contains too much information (seven different simulated signatures), and it has steps and workflows embedded in the figure.
- 3) The two datasets that the authors currently studied are profiled by the same microarray platform (Affymetrix HG-U133 Plus 2.0). It is unclear whether the clusters could be applied to cross-platforms data set. In addition, some of the genes (probes) selected in the clustered represent control probe sets, which might not have any biological meaning in the biological interactions.
- 4) one of the claims made by the authors are the ability of the method in dissecting the underlying gene-gene interactions in the signatures, however, no biological interpretations were provided in the main text of these signatures.

Reviewer #2 (Remarks to the Author):

The manuscript by Grau et al showed a nonlinear data model used to analyze gene expression profiling data, which can be used to discover novel survival signatures and is more discriminating than the 3 signature model initially proposed by Lenz et al. The study is purely silico and did not provide biological and experimental supporting evidences. The survival signatures are based on mRNA levels but not phenotypic protein expression and genomic NGS results.

Comments:

- (1) The simulated interaction concept is unclear in biology. Is it mainly of transcriptional regulations? Or tumor-immune cell interaction? Can some of discovered interaction be validated biologically in literatures?
- (2) Data were trained for discovery in this study. It is hard to tell whether the data regression is meaningful.
- (3) The K27 and K11 signatures represent overlapping opposite survival trends. What's the biological relationship between these two gene signatures? Is there overlapping interaction among

the gene signatures?

(4) The Lenz et al signature is stromal signatures, whereas the new signatures are more of proliferation signatures. A combination of analysis may be more valuable ?

(5) Why Figure 6 a-b have four groups, but Figure c-d have 2 groups, and Figure e-f have 3 groups? How the signatures are used to divide the patients is not well explained.

Reviewer #3 (Remarks to the Author):

Manuscript Title: SDCM: Inference of interactions from high-dimensional datasets and survival in Diffuse Large B-Cell Lymphoma

The Authors Grau et al. in this manuscript reported a novel method (SDCM) to discover gene expression classifiers for patients subtyping, particularly using the diffuse large B-cell lymphoma (DLBCL) gene expression data. The significant association of the classification signature with clinical survival rate was validated by independent cohort. This signature showed improved performance over the previously reported DLBCL gene signatures.

As a method for discrimination of gene expression pattern, SDCM is superior to HC, NMF, FABIA, FABIAS, ICA, as well as PCA which is widely used in genomic classification.

The unique feature of the SDCM model is it uses nonlinear regression to catch the subset of genes (signatures) that may underlie interactions. This is done by an iterative procedure. In every iteration k , a signature is detected. Each iteration applies four steps to discover a signature:

- (1) search for an initial candidate direction using a defined signature function.
- (2) optimize this direction by locally maximizing the signature functional,
- (3) regress a monotonic curve through data points in the weights-based cone and
- (4) selectively dissect the signal parts that are consistent with this curve.

Both the manuscript and the supplemental were well written. The authors also provided the MATLAB code of SDCM for public test and use. I recommend this manuscript to be published on Nature Communications.

Major:

1. In order to construct the initial signature vectors, the initial weights of genes were computed based on the standardized signal (SDCM Supplementary Information, on page 13/68). It is not clear if the standardized signal was computed from Mk -iteratively for each vector or signature, or for the whole matrix M data, or a pre data normalization/standardization is required.
2. It is not surprised that among the total 105 signature, two signals (K27 and k11) were identified to show significantly associated with clinical outcome (Fig 6). It is fair to say that the SDCM is able to capture largely the gene expression heterogeneity among DLBCL, but SDCM signatures are not specific for survival predictions.
3. In the clinical survival association test, both the discovery and validation cohort data were generated from the same microarray array platform HG U133 Plus 2.0. It is not clear what the SDCM's performance is if more validations are conducted on a different platform data such as Illumina beadarray, or RNA-seq data, or using other than DLBCL data.
4. It would be helpful if we could know the biological functions or pathways involved in the top SDCM signatures, particularly in the K27 and K11 that associated with clinical outcomes.
5. Though the authors used GSEA in this study, the GSEA was not mentioned in the Introduction

when the authors argued the “Detecting interactions in GE datasets ...”.

6. Based on the iterative procedure and the selection of the initial candidate direction in each iteration, the SDCM may take computing cost. This was not mentioned or compared to other methods in the text.

Minors:

In the main text, the Figures were not labeled as the order of appearance.

Point-by-point responses to reviewers' comments:

Reviewer #1 (Remarks to the Author):

Grau et al described a novel unsupervised learning concept known as Signal Dissection by Correlation Maximization (SDCM) that dissects large gene expression datasets into signatures. The authors claimed that the derived signature captures a particular signal pattern that was consistently observed in multiple genes and samples, with potential associated with the same underlying genetic interactions. The authors compared their method with other existing clustering methods, and showed that in their method could derived more predictive signatures than other methods in a malignant lymphomas data set. Using another validation set, the authors claimed that their method could derived gene expression signatures that correlated with Progression Free Survival (PFS) of malignant lymphoma in the training set, and similarly showed the same PFS significant in the validation set. Overall, the manuscript is hard to follow as it is very dense and a lot of references to supplementary results. This reviewer has the following comments:

Major comments

1) Unsupervised learning approach is an approach to extract novel gene-gene interactions from high-throughput omics data. However, one of the limitations of the unsupervised approach is the determination of the number of k clusters in a particular data set. The authors have showed that the number of k for a the same dataset (Fig 5), could be extracted with $k=27$ and $k=11$. However, in a realistic application, determining the best (or optimal) number of k is still unclear.

We agree with the reviewer, in any unsupervised signal analysis the number of clusters is hard to determine. If there is a clear gap between signal and noise, SDCM usually performs very well with respect to this problem. This is shown by method validation with diverse simulated datasets having a known true number of contained signatures (please see supplementary figures S1 and S11-S13).

For real data, SDCM identified $\hat{k} = 105$ signatures, i.e. there are no alternative dissections with a different total number of signatures. Stronger and larger discovered signatures tend to be detected first (i.e. they have a lower detection index k ; see supplementary figure S20). Our subsequent analyses after dissection then identified two particular signatures out of the 105 discovered signatures (signature #27 and signature #11) whose expression data are associated with differences in survival of affected patients.

2) Figure 3 is very hard to interpret, as it contains too much information (seven different simulated signatures), and it has steps and workflows embedded in the figure.

We agree that figure 3 contains a lot of information. To improve clarity, we have redesigned the embedded workflow information embedded in figure 3. However, unfortunately we cannot reduce the number of simulated signatures, as a sufficiently complex test signal is required for method validation.

3) The two datasets that the authors currently studied are profiled by the same microarray platform (Affymetrix HG-U133 Plus 2.0). It is unclear whether the clusters could be applied to cross-platforms data set. In addition, some of the genes (probes) selected in the clustered represent control probe sets, which might not have any biological meaning in the biological interactions.

Our key signatures have been validated by independent survival data within the learning cohort ($FDR_{k=27} = 2.21 \cdot 10^{-9}$ and $FDR_{\text{CPHM},k=11} = 2.31 \cdot 10^{-7}$) confirming that our key signatures are of biological origin and not platform-dependent. The fact that some signatures contain contributions from chip-specific control probes (such as AFFX.Bio* probe sets) is a consequence of the complete dissection of the input signal by SDCM. Dissection of such technical signatures is indeed useful as they remove chip-specific offsets (i.e. signal parts that are correlated to control probes) that might otherwise overlap dissected biological signatures. Notably, top probe sets of the two key survival signatures dissected in iterations $k=11$ and $k=27$ do not contain control probe sets, whose names start with “AFFX” on this microarray platform (cf. corresponding columns of Dataset S1 or Supplementary Figs. S26 and S27).

4) one of the claims made by the authors are the ability of the method in dissecting the underlying gene-gene interactions in the signatures, however, no biological interpretations were provided in the main text of these signatures.

The reviewer raises a valid point. As suggested by the reviewer, we have now performed additional analyses of the two discovered signatures predicting survival of DLBCL patients. We performed gene set enrichment analyses for these signatures using a database of 16,513 previously described gene expression signatures. Based on these analyses, we were able to show that $k=27$ is a more refined signature containing genes that might be responsible for the survival difference between ABC and GCB DLBCL. In contrast, the $k=11$ signature revealed a DLBCL subtype that is characterized by a Burkitt lymphoma like expression pattern and low expression of previously identified target genes of the oncogenic nuclear factor-kappa B (NF- κ B) signaling pathway. Interestingly, very recently a novel DLBCL subtype with features of Burkitt lymphoma was identified (Sha, C. *et al.* Molecular High-Grade B-Cell Lymphoma: Defining a Poor-Risk Group That Requires Different Approaches to Therapy. *J. Clin. Oncol.*, 2018). It is tempting to speculate that SDCM identified the same biologic subgroup without using prior knowledge of lymphoma gene expression signatures or survival data. These new findings are presented in the revised results section (on page 14 of the revised manuscript) and are discussed in detail on pages 15 and 16.

Reviewer #2 (Remarks to the Author):

The manuscript by Grau et al showed a nonlinear data model used to analyze gene expression profiling data, which can be used to discover novel survival signatures and is more discriminating than the 3 signature model initially proposed by Lenz et al. The study is purely silico and did not provide biological and experimental supporting evidences. The survival signatures are based on mRNA levels but not phenotypic protein expression and genomic NGS results.

Comments:

(1) The simulated interaction concept is unclear in biology. Is it mainly of transcriptional regulations? Or tumor-immune cell interaction? Can some of discovered interaction be validated biologically in literatures?

SDCM is a generic method to discover correlations in data. There is no direct link to or any specific assumption about underlying interactions. Correspondingly, SDCM is applicable to many different data sets and the identified correlations can arise from very different interactions depending on the nature of the analyzed data. We have now clarified this in the last paragraph of the introduction (page 3). For example, for the analyzed gene expression data one expects

that the discovered correlations reflect transcriptional regulation. However, the results of an SDCM analysis are sets of connected entities identified by correlations and not by specific interactions. In this sense, SDCM is similar to other methods such as principal components analysis (PCA). Like principal components obtained by PCA, results of an SDCM analysis can however provide a basis to identify relevant interactions and to focus subsequent biological research.

With respect to validating discovered interactions in the existing literature, we have already confirmed that we rediscovered three previously published DLBCL signatures, including stromal signatures from ref. 2 (Supplementary Fig. S17-S19), despite not using any survival data for learning as originally done. As detailed below in comment (3), we have now detected two additional signatures predicting survival of DLBCL patients.

(2) Data were trained for discovery in this study. It is hard to tell whether the data regression is meaningful.

The reviewer raises an important point. We purposely withheld all clinical patient metadata from signature discovery in the learning DLBCL cohort. Thus, the significant association of signatures detected in iteration $k=11$ and $k=27$ with patient survival is an independent validation of their biological origin, indirectly confirming our data regression model. This analysis concept is comparable to the unsupervised analysis that originally identified the molecular DLBCL subtypes ABC DLBCL and GCB DLBCL (11, Alizadeh 2000). Additionally, we showed in an independent DLBCL cohort that the same correlations exist for the gene ordering inferred from data regression in the learning cohort. Likewise, the same survival differences are observed for patients in the validation cohort. In addition, we were now able to link discovered signatures to a clinically relevant subgroup of DLBCL (see next point).

The data regression is based on the following idea: The expression data of two genes has maximum correlation if they are proportional to each other in many samples, i.e. if there is a linear relationship between the expression data of the two genes. We now generalize this linear relationship to monotonic relations (that do not need to be linear). Thus, by going from one sample to another, the gene expression data of both genes only need to change in the same direction (i.e. increase or decrease) but their ratio is allowed to change. In this way we can adapt and represent more precisely a much larger class of signal patterns. This was clarified in the first results paragraph (on page 3 in the revised manuscript).

(3) The K27 and K11 signatures represent overlapping opposite survival trends. What's the biological relationship between these two gene signatures? Is there overlapping interaction among the gene signatures?

As outlined above in answer to reviewer 1, we have now performed additional analyses of the two discovered signatures predicting survival of DLBCL patients. We performed gene set enrichment analyses for these signatures using a database of 16,513 previously described gene expression signatures. Based on these analyses, we were able to show that $k=27$ is a more refined signature containing genes that might be responsible for the survival difference between ABC and GCB DLBCL. In contrast, the $k=11$ signature revealed a DLBCL subtype that is characterized by a Burkitt lymphoma like expression pattern and low expression of previously identified target genes of the oncogenic nuclear factor-kappa B (NF- κ B) signaling pathway. Interestingly, very recently a novel DLBCL subtype with features of Burkitt lymphoma was identified (Sha, C. *et al.* Molecular High-Grade B-Cell Lymphoma: Defining a Poor-Risk Group That Requires Different Approaches to Therapy. *J. Clin. Oncol.*, 2018). It is tempting to speculate that SDCM identified the same biologic subgroup without using prior knowledge of

lymphoma gene expression signatures or survival data. These new findings are presented in the revised results section (on page 14 of the revised manuscript) and are discussed in detail on pages 15 and 16.

(4) The Lenz et al signature is stromal signatures, whereas the new signatures are more of proliferation signatures. A combination of analysis may be more valuable ?

We agree with the reviewer and have included this combination in our unbiased construction of survival models based on all SDCM signatures and all signatures from comparison methods. Thus, both rediscovered stromal signatures (supplementary Figures 17 and 18) had the same chance to be selected as part of constructed bi- and tri-variate survival models as all other signatures. However, this analysis revealed that other combinations were significantly better at survival prediction. This analysis has been clarified in the first paragraph of the results section “Survival models and method comparison” (on page 11 in the revised manuscript).

In addition, we have investigated to include the previously published survival predictor score (that uses both stromal signatures and the germinal center B-cell signature) as additional information source on top of our identified 2-signature model (Supplementary Note 2.9). But a likelihood ratio test (LRT) showed that this score did not contribute any significant additional information. More precisely, p_{LRT} was 0.07, so there may be a weak trend towards borderline significance. In contrast, testing our 2-signature model on top of the old 3-signature predictor score yielded significant additional information with $p_{LRT}=1.3e-8$. These results were stated in the last paragraph of results section “Survival models and method comparison” (on page 13 of the revised manuscript).

In summary, while there is a weak trend towards even larger survival differences, the difference is not statistically significant and not strong enough to justify going from a bivariate to a tri-variate model, as the possible loss of robustness and generalizability when transferring the model to other DLBCL cohorts is more influential in our view. From a biological point of view, we interpret the data that the refined cell-of-origin signature ($k=27$) and the Burkitt lymphoma like NF- κ B-related signature ($k=11$) are the predominant factors that determine patient outcome of affected patients, while the stromal signatures seem to be less important.

(5) Why Figure 6 a-b have four groups, but Figure c-d have 2 groups, and Figure e-f have 3 groups? How the signatures are used to divide the patients is not well explained.

As we have shown by a bivariate Cox proportional hazard model, expression of signatures $k=27$ and $k=11$ is significantly associated with a survival difference in DLBCL patients ($p_{CPHM,k=27} = 2.1 \cdot 10^{-11}$ and $p_{CPHM,k=11} = 2.2 \cdot 10^{-9}$). To visualize this trend with established Kaplan Meier survival curves, one has to introduce formal cutoffs for partitioning the patient cohort. For a continuous trend, no ideal cutoffs exist. To visualize the spread from low to high survival in the full DLBCL cohort, we chose cohort partitions according to the patient risks known from the Cox proportional hazard model. More precisely, patients having a risk $>175\%$ of the average risk, patients between 100% and 175% , between 100% and $1/175\%$ and $<1/175\%$ are shown in separate survival curves. Log-rank tests between these curves confirm the significance of the Cox proportional hazard model. However, the main point is that a similar survival spectrum is visible in the validation cohort (using the same signatures and the Cox proportional hazard model from the learning cohort, of course without any refitting). Within GCB DLBCL, there is no continuous trend, but the upper quartile of predicted risks according to our model is shown to have significantly lower average survival than the other three quartiles of GCB DLBCL patients. Again, not the cutoff position, but the fact that likewise

differences are observed in the validation cohort is the main statement of this figure, as this indicates that these patients could profit from an adapted therapy and that this is a characteristic of DLBCL and not just a single DLBCL cohort. This has been clarified in the in the second paragraph of results section “Survival models and method comparison” (on pages 12 and 13 in the revised manuscript). Finally, we have depicted the survival trends revealed by our molecular predictor within clinical risk classes by IPI. Initially, we also tried to visualize the trend here via a partitioning in four curves. However, the number of patients having IPI was only sufficient for partitioning into terciles (and not quartiles). We have clarified the caption of this figure with respect to the chosen visualization of discovered survival trends (on page 13 of the revised manuscript).

Reviewer #3 (Remarks to the Author):

Manuscript Title: SDCM: Inference of interactions from high dimensional datasets and survival in Diffuse Large B-Cell Lymphoma

The Authors Grau et al. in this manuscript reported a novel method (SDCM) to discover gene expression classifiers for patients subtyping, particularly using the diffuse large B-cell lymphoma (DLBCL) gene expression data. The significant association of the classification signature with clinical survival rate was validated by independent cohort. This signature showed improved performance over the previously reported DLBCL gene signatures.

As a method for discrimination of gene expression pattern, SDCM is superior to HC, NMF, FABIA, FABIAS, ICA, as well as PCA which is widely used in genomic classification.

The unique feature of the SDCM model is it uses nonlinear regression to catch the subset of genes (signatures) that may underlie interactions. This is done by an iterative procedure. In every iteration k , a signature is detected. Each iteration applies four steps to discover a signature:

- (1) search for an initial candidate direction using a defined signature function.
- (2) optimize this direction by locally maximizing the signature functional,
- (3) regress a monotonic curve through data points in the weights-based cone and
- (4) selectively dissect the signal parts that are consistent with this curve.

Both the manuscript and the supplemental were well written. The authors also provided the MATLAB code of SDCM for public test and use. I recommend this manuscript to be published on Nature Communications

Major:

1. In order to construct the initial signature vectors, the initial weights of genes were computed based on the standardized signal (SDCM Supplementary Information, on page 13/68). It is not clear if the standardized signal was computed from M_k -iteratively for each vector or signature, or for the whole matrix M data, or a pre data normalization/standardization is required.

SDCM takes the input signal matrix M_0 that may or may not have undergone external data preprocessing, such as \log_2 –transformation or median centering of raw gene expression intensities. All M_k have the same units as M_0 , i.e. the input data is never transformed. Dissection of discovered signature matrices E_k also retains the input units.

The standardized signal is only needed for searching initial candidate directions at each iteration k . Here, correlations to gene and sample axes are not available and a replacement is needed to get the search and signature focus started. To compute the initial weights for candidates, we compute the standardized version M_{k-1}^{δ} of the whole matrix M_{k-1} , at the beginning of each iteration k .

This standardization increases robustness against data outliers. Moreover, the rationale of this standardization is that absolute values of standardized vector components are larger for signatures having their signal contributions concentrated in fewer genes, and lower for signal parts distributed over many genes, which is more typical for sources of noise.

Pre-focusing with these initial weights is performed conservatively, as its main purpose is to exclude noise dimensions. For example, a signature affecting only 10/20000 of measured genes is harder to detect, if all gene weights are equal, compared to weighting down the 19990 genes not having any strong signal in signature direction. Computation of M_{k-1}^{δ} itself is also iterative via a convergence loop, as described in SI 1.12.1. We thank the reviewer for pointing this out. This has been clarified on page 13/69 of the revised supplement.

2. It is not surprised that among the total 105 signature, two signals (K27 and k11) were identified to show significantly associated with clinical outcome (Fig 6). It is fair to say that the SDCM is able to capture largely the gene expression heterogeneity among DLBCL, but SDCM signatures are not specific for survival predictions.

We agree that SDCM signatures per se are not necessarily specific to survival prediction. In particular, not only the biological heterogeneity in DLBCL, but also any overlapping technical or laboratory effect is dissected as signature until the remaining signal appears to consist of only leftover noise (complete signal dissection). This is why we used survival follow-up data only after signature dissection to evaluate signatures for their biological relevance and specificity. We did the same for described comparison methods. The SDCM signatures at indices $k=27$ and $k=11$ predicted survival better than any signatures from any other unsupervised method (see reported relative likelihoods in Table S4). In this sense, SDCM is able to extract the most survival-specific signatures. More precisely, these signatures have survival associations with reported p values of $p_{\text{CPHM},k=27} = 2.1 \cdot 10^{-11}$ and $p_{\text{CPHM},k=11} = 2.2 \cdot 10^{-9}$ in the Cox proportional hazard model (CPHM). Even with the most conservative Bonferroni correction, corresponding false discovery rates for these two signatures in the family of 105 SDCM signatures are $FDR_{k=27} = 2.21 \cdot 10^{-9}$ and $FDR_{\text{CPHM},k=11} = 2.31 \cdot 10^{-7}$, respectively. So, while one would expect 5-6 signatures of the 105 to show survival associations, if one used a typical 5% alpha error, none would be expected setting the alpha error threshold to 10^{-7} .

We believe this performance regarding detection specificity is due to the ability of SDCM to dissect overlapping signatures. Notably, supervised survival analyses that directly associate each gene with the same survival follow-up data, could not reveal these synergistic and anti-aligned survival trends of the two signatures, as such analyses usually only use the complete signal sum per gene and not dissected signature-specific signal summands.

For clarification, we no longer state survival specificity for SDCM signatures generically (in the first paragraph of discussion on page 14 of the revised manuscript), but only for the two signatures for which this has been shown.

3. In the clinical survival association test, both the discovery and validation cohort data were generated from the same microarray array platform HG U133 Plus 2.0. It is not clear what the

SDCM's performance is if more validations are conducted on a different platform data such as Illumina beadarray, or RNA-seq data, or using other than DLBCL data.

Transferring a signature to another technological platform is indeed associated with difficulties. For example, when measuring the same biological setting with different probe affinities or different sample labeling protocols, technological scalings or offsets on top of true biological signals are to be expected. To be able to transfer a signature two requirements have to be met:

- (i) the signature is defined by correlation patterns that are exclusively of biological origin
- (ii) the target signal has already been cleaned from any technological signal parts specific for the target platform.

We feel that (i) is met, i.e. the identified signatures are biologically meaningful, as we did not use survival data to train signatures and the strong association of these signatures with survival data provides (as explained above in answer to comment #2) an independent validation of the biological origin of discovered signatures $k=27$ and $k=11$ in the learning cohort. It is unlikely that any technological properties of the microarray platform would have such strong survival association by pure chance. However, (ii) requires dissections of large cohorts measured with the target platform to identify and then remove technological signatures from the signal. So far, we have not addressed this issue mainly as of lack of sufficiently large DLBCL datasets. This should be addressed in the near future, but these analyses are beyond the scope of the current manuscript.

4. It would be helpful if we could know the biological functions or pathways involved in the top SDCM signatures, particularly in the K27 and K11 that associated with clinical outcomes.

As outlined above in answer to reviewer 1 and 2, we have now performed additional analyses of the two discovered signatures predicting survival of DLBCL patients. We performed gene set enrichment analyses for these signatures using a database of 16,513 previously described gene expression signatures. Based on these analyses, we were able to show that $k=27$ is a more refined signature containing genes that might be responsible for the survival difference between ABC and GCB DLBCL. In contrast, the $k=11$ signature revealed a DLBCL subtype that is characterized by a Burkitt lymphoma like expression pattern and low expression of previously identified target genes of the oncogenic nuclear factor-kappa B (NF- κ B) signaling pathway. Interestingly, very recently a novel DLBCL subtype with features of Burkitt lymphoma was identified (Sha, C. *et al.* Molecular High-Grade B-Cell Lymphoma: Defining a Poor-Risk Group That Requires Different Approaches to Therapy. *J. Clin. Oncol.*, 2018). It is tempting to speculate that SDCM identified the same biologic subgroup without using prior knowledge of lymphoma gene expression signatures or survival data. These new findings are presented in the revised results section (on page 14 of the revised manuscript) and are discussed in detail on pages 15 and 16.

5. Though the authors used GSEA in this study, the GSEA was not mentioned in the Introduction when the authors argued the "Detecting interactions in GE datasets ...".

We thank the referee for pointing this out. We have now mentioned GSEA in this paragraph (on page 2 of the revised manuscript) and have clarified at the end of the introduction (on page 3) that we only compare SDCM to other unsupervised learning methods.

6. Based on the iterative procedure and the selection of the initial candidate direction in each

iteration, the SDCM may take computing cost. This was not mentioned or compared to other methods in the text.

We have analyzed the asymptotic complexity of the SDCM algorithm, both analytically and numerically. Due to space restrictions, we had to move details into the SI (Supplementary Note 1.11). In the main text, we only report the resulting analytical complexity $O(\hat{k}(m+n)nm)$. Additionally, we have shown that measured computation times scaled quadratically in the genes count m (Supplementary Fig. S14). For practical purposes, we also report that the average runtime is about 5 minutes per signature for a dataset with $m = 20000$ genes and $n = 100$ samples.

Minors:

In the main text, the Figures were not labeled as the order of appearance.

The figures are now labeled in the order of appearance in the revised manuscript.

Once again, we would like to thank the editors and the reviewers for the thorough evaluation of our manuscript.

Reviewers' comments:

Reviewer #1 (Remarks to the Author):

In this revised manuscript, the authors have done a good job in addressing the previous comments and improved the presentation of results.

Reviewer #2 (Remarks to the Author):

This reviewer has question for the clinical significance of this study. For example, the signatures were generated from correlations in large numbers of patients. How could they be applied in the clinic when single patient's result was given and lack of biological study?

(1) The author only mentioned that the $K = 11$ signature is similar with the newly identified MHG signature, but did not specify.

(2) As addressed in the second point by the author, the signatures consist of genes that are significantly associated with each other. Then how to tell which gene is the driver for lymphomagenesis or poor clinical outcomes?

(3) In page 12, the manuscript said "Neither was it predictive on top of the rediscovered COO signature $k = 12$ ". Then why in the addressed 3rd point, the author said $K=27$ signature is the refined signature containing genes differentiating survival between ABC and GCB DLBCL?

(4) In page 10, why the gene signature is disease unspecific?

(5) In the fifth point addressed, the author mentioned that the cutoffs were chosen by survival groups known from the Cox proportional hazard model. Was this method only done for cohort 1 only or also in the validation cohort? Are the cutoffs in two cohorts same? As the author said "no ideal cutoffs exist", then how to apply this method to prospective patients?

(6) What do the four lines in Figure 6a-b mean? 2 lines each for Signature K11 and K27? How the GCB subtype was defined in Figure c-d? (the author said "Neither was it predictive on top of the rediscovered COO signature $k = 12$ ")

(7) What are the usefulness of $K=6$ and $K=12$ signatures, which are also mentioned in the manuscript? Other minor issues include grammars such as "patient samples" in page 10.

Reviewer #3 (Remarks to the Author):

The authors have addressed all my comments.

The SDCM is technically sound. Cross platform testing is always a challenging and vital task in gene signature development and method validation. It would be helpful if the authors could present any results using DLBCL RNA-seq data from TCGA.

To compromise, the authors may also try to use any other cancer types of RNA-seq data that are most commonly available nowadays to explore the usefulness and performance of the SDCM method as a generic method for gene signature discovery, since this paper is mostly methodology, not specifically about DLBCL.

Point-by-point responses to reviewers' comments:

Reviewer #1 (Remarks to the Author):

In this revised manuscript, the authors have done a good job in addressing the previous comments and improved the presentation of results.

We would like to thank the reviewer again for his helpful suggestions and comments.

Reviewer #2 (Remarks to the Author):

This reviewer has question for the clinical significance of this study. For example, the signatures were generated from correlations in large numbers of patients. How could they be applied in the clinic when single patient's result was given and lack of biological study?

Our study provides a generic method of unsupervised learning. Using the context of DLBCL as example, we have shown that SDCM is able to discover signatures revealing strong and novel survival differences, without utilizing survival data to discover these signatures. (Just GE data of the detection cohort is input to SDCM; follow-up data is only used later for signature validation and construction of a predictor based on SDCM signatures.) With the new RNA-sequencing based validation cohort, we now have also shown that discovered signatures are sufficiently robust and specific to maintain their significant predictive capabilities across technological borders. Especially within GCB DLBCL, we have identified a subgroup of patients with significantly inferior survival compared to other GCB DLBCL patients (Supplementary Fig. S33). This might have potential impact on therapy decisions in the future. In contrast to previously defined ABC and GCB DLBCL subtypes that showed a strong variation in survival difference when crossing cohort borders, our predictor generalizes more robustly across cohorts (Supplementary Fig. S31). Hence, top genes in our signatures provide rational targets for further research on the functional significance of DLBCL-specific driver genes. Even though our method provides a generic means to identify such targets, the focus of this study is the method itself and not the analysis of potential cancer targets.

With respect to prospective clinical applications to single patient's results, one has to separate methodological and technological aspects. Mathematically, it is straightforward to apply our SDCM signatures to any new gene expression data for a single patient and then utilize our predictor model to obtain an individual risk estimate for this patient, provided that gene expression values have been normalized relative to average expression in DLBCL. However, the latter is technologically problematic, especially for microarrays (for example, due to superposed laboratory effects by probe affinities, labeling differences etc.). Typically, no normalization/calibration baseline measurements are available alongside the individual measurement. With gene expression intensities based on a RNA-sequencing measurements that are not affected by probe affinities, this single-patient application becomes potentially possible, but still needs good preprocessing and normalization of the new single patient's measurement to merge it into the available RNA-sequencing cohort and obtain $\log_2(\text{ratio})$ s for each gene. Once this is achieved, our predictor can be applied. However, this technological calibration needed for standalone patient measurements is out of the scope of this study.

(1) The author only mentioned that the $K = 11$ signature is similar with the newly identified MHG signature, but did not specify.

We have shown that both correlated and anti-correlated genes of $k = 11$ were significantly enriched with signatures differentiating Burkitt lymphoma (BL) from DLBCL samples in a supervised analysis ("HUMMEL_BURKITTES_LYMPHOMA_DN" with enrichment score $es = -0.903$; "HUMMEL_BURKITTES_LYMPHOMA_UP" with $es = 0.741$; each with $p \leq 0.001$ by permutation tests). Hence, samples

with high strength in signature $k = 11$ showed a BL-like gene expression pattern. This could be a potential link to MHG as can be seen from the comparison of the average expression level of the BL signature between MHG and GCB DLBCL at the bottom of Fig. 3 in the manuscript by Sha et al.⁴⁶ and the super-linear increase in sample strengths in signature $k = 11$ on the right side of each of the three cohorts in Fig. 5b and Supplementary Fig. S27. Inferior survival and size of the identified high risk subgroup within GCB DLBCL (cf. Supplementary Fig. S33) provide further support of the potential link to MGH. However, as this is indirect evidence we have toned down our statements regarding the relationship to MGH in the revised version of the manuscript on page 16.

(2) As addressed in the second point by the author, the signatures consist of genes that are significantly associated with each other. Then how to tell which gene is the driver for lymphomagenesis or poor clinical outcomes?

The intention of our novel SDCM method is to identify relevant gene targets for further research. These genes are detected based on gene-gene correlations in form of signature top genes. Certain identification of individual driver genes is not possible based on gene expression data alone. Ultimately, biological identification of driver genes will require comprehensive validation experiments with functional in vitro and in vivo models of DLBCL. However, this is out of the scope of this manuscript. In the revised manuscript we have now clarified the aim and goals of SDCM on page 15 in the revised "Discussion".

(3) In page 12, the manuscript said "Neither was it predictive on top of the rediscovered COO signature $k = 12$ ". Then why in the addressed 3rd point, the author said $K=27$ signature is the refined signature containing genes differentiating survival between ABC and GCB DLBCL?

Multiple overlapping signatures are able to sort patients by previously defined subtypes ABC and GCB DLBCL, but these signatures are only partially correlated to each other and not biologically identical. The original subtype signature¹² was not defined by maximizing inner correlation between its member genes, but by a hierarchical clustering. Instead of clustering the gene expression signal, SDCM models it as a sum of potentially *overlapping* signatures, each with maximized inner correlation. Our results indicate that the original ABC/GCB DLBCL signature represents multiple partially correlated but distinct biological contributions to the overall gene expression signal, i.e. it is biologically less specific and the signal is not fully dissected. We termed signature $k=12$ the rediscovered COO signature, as it had the highest gene set enrichment for the previously known ABC/GCB DLBCL signatures (Supplementary Fig. S19b-c). While signatures $k=27$ and $k=11$ also roughly sorted samples from patients by their subtype, their gene strength and associated top genes are farther away from these original ABC/GCB DLBCL gene signatures.

Signature $k=11$ was not predictive on top of $k=12$, but highly synergistic in survival prediction when combined with $k=27$. This underlines the necessity to dissect these overlapping but biologically unequal gene expression programs precisely.

From an analytical/geometrical point of view, these three high-dimensional signatures $k=11$, 12 and 27 can be thought to correspond to the red, green and blue signatures in our 3D concept example. These are partially correlated in the direction given by the first principal component in Fig. 1i, but they are still far from identical. The original COO signature roughly corresponds to this principal component in the empty gene space between three signatures, but signature $k=12$ was the nearest to previous GCB/ABC DLBCL gene signatures in terms of gene set enrichment, hence the naming.

Assuming that the cell of origin is indeed the biological correct explanation of the primary part of survival differences in DLBCL, and given our results on survival level in now three large DLBCL cohorts including a RNA-sequencing cohort, we feel that $k=27$ provides a refined distinction for the cell of origin. I.e., while signature $k=12$ may be closer to previous ABC/GCB DLBCL signatures in terms of gene set enrichment, it

did not represent the strongest survival trend, which we think is still caused by the cell of origin. In this sense, $k=27$ is a refined signature for a more robust ordering of samples by their COO. We have clarified this in the revised manuscript on pages 11 and 16.

(4) In page 10, why the gene signature is disease unspecific?

This statement refers to signature $k=6$ that reproduced the patients' genders almost by 100% in all three cohorts, but as expected did not show any strong association to either subtype or survival (see Supplementary Fig. S16). From the perspective of method validation, this signature is a proof of concept, as gender was no input data to SDCM. It can also serve as a practical control signature. For example, it confirmed consistency of our RNA-sequencing analysis and provided sample annotations for the added validation, as this clear gender association was again reproduced (see Supplementary Fig. S16). However, as we cannot conclude from these data that DLBCL is gender-unspecific, we reduced the related statement in the manuscript to "as disease-unspecific as gender". We thank the reviewer for pointing this out.

(5) In the fifth point addressed, the author mentioned that the cutoffs were chosen by survival groups known from the Cox proportional hazard model. Was this method only done for cohort 1 only or also in the validation cohort? Are the cutoffs in two cohorts same? As the author said "no ideal cutoffs exist", then how to apply this method to prospective patients?

The Cox proportional hazard predictor computes for each sample a risk value in units of average risk in the detection/learning cohort. Only for visualization of this model by Kaplan-Meier curves, risk cut points were introduced to group patients by their predicted molecular risks. Identical cuts were applied to all cohorts when visualizing their inner survival differences; see also reply to point 6 below.

The predictor model is independent of these cut points. Provided that genes in underlying signatures are measured and properly normalized, our model can provide a risk prediction for any gene expression measurement of a prospective patient. The same risk prediction has been computed for all patients in the add-on RNA-sequencing-based validation cohort. Again, same risk cuts were applied to visualize overall survival differences within this cohort (new Supplementary Fig. S33; no progression-free survival data available for this cohort). This was clarified in the revised legend of Fig. 6. on page 13.

(6) What do the four lines in Figure 6a-b mean? 2 lines each for Signature K11 and K27? How the GCB subtype was defined in Figure c-d? (the author said "Neither was it predictive on top of the rediscovered COO signature $k = 12$ ")

Both Fig. 6a (for the detection cohort) and 6b (for the microarray-based validation cohort) visualize the results of our 2-signature predictor model that is based on both signatures $k=27$ and $k=11$ simultaneously (in form of a bivariate Cox proportional hazard model). Each sample is assigned a risk by this model, based on its sample strengths in these two signatures. This model is continuous and to visualize it by usual Kaplan-Meier survival curves, one has to group samples. We show them in four groups, defined by cut points of their predicted molecular survival risks. The lowest survival curves show patients predicted to have a molecular risk of $\geq 175\%$ the average risk in the detection cohort, the highest curve only those with a risk of $\leq 1/175\%$. To obtain four groups, another cut point was introduced at a predicted risk of 100%, i.e. exactly at average risk as learned from the detection cohort. In Figure 6c-d, we show survival differences predicted by our signatures within GCB DLBCL. To this end, we utilized the previous DLBCL classifications from the respective dataset. We have clarified this point in the amended legend of Fig. 6 on page 13 of the revised manuscript.

(7) What are the usefulness of K=6 and K=12 signatures, which are also mentioned in the manuscript? Other minor issues include grammars such as “patient samples” in page 10.

Signature k=6 is an analytical proof of concept for SDCM and may serve as control signature for processing and annotation pipelines in practice, as it is strongly associated with gender. We have amended Supplementary Fig. S16 to confirm this claim in the added RNA-sequencing based validation cohort. Signature k=12 was the closest to previously known ABC and GCB DLBCL related signatures in terms of signature enrichment. It also serves the purpose of explaining that the previously known COO effect can be dissected into overlapping signatures of high inner correlation (k=11, 12, 27) that show distinct biological properties, in particular distinct associations to patient survival. We thank the reviewer for pointing out this valid point. We have modified the revised manuscript accordingly on pages 11 and 16.

Reviewer #3 (Remarks to the Author):

The authors have addressed all my comments.

The SDCM is technically sound. Cross platform testing is always a challenging and vital task in gene signature development and method validation. It would be helpful if the authors could present any results using DLBCL RNA-seq data from TCGA.

To compromise, the authors may also try to use any other cancer types of RNA-seq data that are most commonly available nowadays to explore the usefulness and performance of the SDCM method as a generic method for gene signature discovery, since this paper is mostly methodology, not specifically about DLBCL.

The reviewer raises a valid point. As suggested by the reviewer, we have performed comprehensive cross-platform testing. We obtained access to the currently largest RNA-sequencing cohort of DLBCL samples³⁹. We aligned raw read files for all available 624/775 DLBCL samples in the RNA-sequencing core set of this study against the current coding RefSeq hg38 transcriptome (Supplementary Note 2.4). To compare with the microarray-based data, we have mapped Affymetrix U133 probe sets and aggregated RNA-sequencing reads on gene level. Not all microarray probe sets had a match (depicted as gray lines in heatmaps for this new validation cohort, see Fig. 5).

Despite bridging across platforms and going from progression-free survival (PFS) to overall survival (as no PFS data was available for the RNA-sequencing cohort), results validated discovered SDCM signatures on two levels. First, we projected each RNA-sequencing sample onto the learned signature gene axis and then sorted all patients in this second validation cohort by their signature strengths (cf. Supplementary Note 2.4). Application to both signatures “k=27” and “k=11” yielded heatmaps as shown in Supplementary Fig. S26 and Fig. S27. As can be seen, signature genes again showed high co-regulation for RNA-sequencing based gene expression.

Additional to this validation of biological correlations on gene expression level, we again associated sample strengths in these two signatures with independent patient survival data according to our 2-signature predictor model. Significant differences in patient survival remained, in the whole cohort and in particular within the GCB DLBCL sub-cohort (Supplementary Fig. S33).

These findings strongly support our hypothesis that our unsupervised learning concept, i.e. dissection of gene expression data in the detection cohort without using any survival data, is able to produce robust signatures with good generalization properties across cohorts. This holds true especially compared with

the generalizability of survival differences between ABC DLBCL and GCB DLBCL patients in the same three cohorts (compare Supplementary Fig. S32).

Again we would like to thank the editors and all reviewers for the comprehensive evaluation of our study and their very helpful suggestions.

REVIEWERS' COMMENTS:

Reviewer #3 (Remarks to the Author):

In this article, Dr Grau et al presented a new model to dissect features from high-dimensional gene expression data and demonstrated its usefulness value by an application in DLBCL gene signature discovery. I appreciate the authors have taken great efforts and have addressed the validation questions.

In the validation tests, the signatures were transferred to new data sets using the same gene axis and gene weights. It will be helpful if the authors could provide a supplemental Table that lists the gene names, the weight of each genes, each data sets (I assume the gene weight is different in microarray data and RNA-seq data), particularly for signature k27, k11, k12.

I assume the heatmap in Fig 5 was made by weighted expression values of the three data sets. The Kaplan-Meier survival estimate in Fig. 6 showed different survival groups. Did the authors use the weighted gene signatures to stratify (by clustering) the patients samples into different survival groups, or is there a weight score for each patients for grouping.

Point-by-point responses to reviewers' comments:

Reviewer #3 (Remarks to the Author):

In this article, Dr Grau et al presented a new model to dissect features from high-dimensional gene expression data and demonstrated its usefulness value by an application in DLBCL gene signature discovery. I appreciate the authors have taken great efforts and have addressed the validation questions.

In the validation tests, the signatures were transferred to new data sets using the same gene axis and gene weights. It will be helpful if the authors could provide a supplemental Table that lists the gene names, the weight of each genes, each data sets (I assume the gene weight is different in microarray data and RNA-seq data), particularly for signature k27, k11, k12.

We thank the referee for this valuable suggestion. We have now added columns for gene weights $\{w^g\}$ for all 105 discovered signatures (including $k = 27$, $k = 11$ and $k = 12$) in Supplementary Data 1. This table also lists Affymetrix probeset IDs, associated gene names, final gene axes $\{b_l^g\}$ correlations $\{r^g\}$ to them, $\{p^g\}$ values for these correlations and gene strengths $\{u^g\}$. Alternatively, gene weights can also be computed from $\{r^g\}$ and $\{p^g\}$ (see signature focus in methods).

However, gene weights are not different for each technology, but are an inherent property of the discovered signature (like the gene axis). For the transfer of the signature to another cohort, a mapping between source probe sets and measured genes or transcripts by the target technology has to be performed. Then, samples of the target cohort are projected on the mapped but otherwise unchanged discovered gene axis, using unchanged gene weights. This yields sample strength values, giving rise to an ordering of samples in the target cohort by the discovered signature. Subsequent analyses such as association with survival are then based on these sample strengths.

I assume the heatmap in Fig 5 was made by weighted expression values of the three data sets. The Kaplan-Meier survival estimate in Fig. 6 showed different survival groups. Did the authors use the weighted gene signatures to stratify (by clustering) the patients samples into different survival groups, or is there a weight score for each patients for grouping.

The partitioning of sample data in Fig. 6 into different survival groups has only been done to improve the presentation of the underlying continuous survival model. More specifically, based on sample strengths in the detection cohort for the two discovered survival signatures, we fitted a bivariate Cox proportional hazard model (CPHM) to corresponding survival metadata in the detection cohort. This model was then applied to sample strengths for the same two signatures in the two independent validation cohorts. The CPHM then predicts a risk value for each patient sample. Values for all samples of each cohort show a continuous risk spectrum. Thus, there is no clustering involved. Only for the final presentation of our CPHM and its predicted risk values, we show Kaplan-Meier survival curves for chosen cutoffs of $\geq 175\%$ times the average risk (in the detection cohort), $\geq 100\%$ and $\geq 1/175\%$, resulting in the four curves that visualize discovered survival differences. For direct visual comparison, identical risk cutoff values have been used for presentation of survival differences in validation cohorts.

Heatmaps in Fig. 5 show unweighted $\log_2(\text{ratios})$, separately for each cohort. Gene weights are only used here to select the top 40 genes in the detection cohort for presentation of the two signatures. We have amended the figure caption to clarify this.